# PrismLayers: Open Data for High-Quality Multi-Layer Transparent Image Generative Models

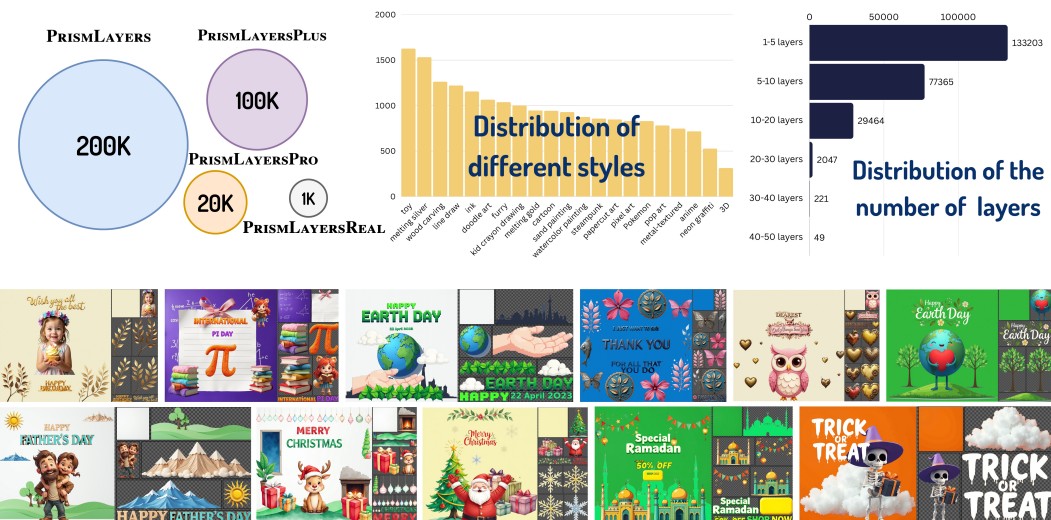

Figure 1: Illustration of key statistics from PrismLayers (number of layers) and PrismLayersPro (different styles), along with representative high-quality synthetic multi-layer transparent images from PrismLayersPro.

## Abstract

Generating high-quality, multi-layer transparent images from text prompts can unlock a new level of creative control, allowing users to edit each layer as effortlessly as editing text outputs from LLMs. However, the development of multi-layer generative models lags behind that of conventional text-to-image models due to the absence of a large, high-quality corpus of multi-layer transparent data. We address this fundamental challenge by: (i) releasing four open, ultra-high-fidelity datasets—PrismLayers, PrismLayersPlus, PrismLayersPro, and PrismLayersReal —consisting of 200K, 100K, 20K, and 1K multi-layer transparent images with accurate alpha mattes, respectively. (ii) introducing a training-free synthesis pipeline that generates such data on demand using off-the-shelf diffusion models, and (iii) delivering a strong multi-layer generation model, ART+, which matches the aesthetics of modern text-to-image generation models. The key technical contributions include: LayerFLUX, which excels at generating high-quality single transparent layers with accurate alpha mattes, and MultiLayerFLUX, which composes multiple LayerFLUX outputs into complete images, guided by human-annotated semantic layout. To ensure higher quality, we apply a rigorous filtering stage to remove artifacts and semantic mismatches, followed by human selection. Fine-tuning the state-of-the-art ART model on our synthetic PrismLayersPro yields ART+, which outperforms the original ART in 60% of head-to-head user study comparisons and even matches the visual quality of images generated by the FLUX.1-[dev] model. Our work establishes a solid dataset foundation for multi-layer transparent image generation, enabling research and applications that require precise, editable, and visually compelling layered imagery.

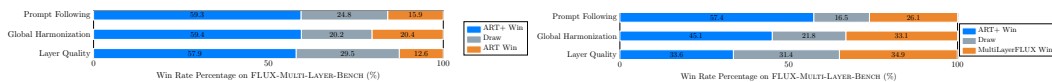

Figure 2: User study results on the effectiveness of PRISMLAYERSPRO. Left: ART+ v.s. ART. Right: ART+ v.s. MultiLayerFLUX. With fine-tuning on PRISMLAYERSPRO, ART+ achieves the best performance.

# 1 INTRODUCTION

Despite remarkable advances in text-to-image diffusion models, users still face significant challenges in refining outputs to achieve satisfactory results. The difficulty lies in the fact that users cannot precisely articulate their visual requirements before seeing generated images, leading to laborious post-processing workflows. The fundamental issue here is that existing diffusion models are designed to produce single-layer images, lacking the transparent layers and precise alpha mattes required for flexible, layer-wise editing. Modern image editing workflows rely on multi-layered structures for the smooth adjustment of individual elements without causing disruption to the entire composition.

In this paper, we argue for a paradigm shift—from text-to-image generation to text-to-layered-image generation. Such an evolution would empower models to support flexible, layer-wise editing operations that align closely with professional design workflows. The fundamental challenge hindering progress in this area is the lack of high-quality multi-layer image datasets featuring both visually appealing transparency and accurate alpha mattes. Bridging this gap is essential to unlocking the full potential of layered image generation with diffusion models.

Nevertheless, existing literature still relies on the conventional pipeline of fine-tuning generative models on limited, low-quality crawled multi-layer datasets. These datasets have two major drawbacks: (i) aesthetic quality: our empirical analysis shows that the aesthetic scores of crawled multi-layer images are significantly lower than those of RGB images generated by state-of-the-art diffusion models like FLUX.1-[dev]. As a result, we empirically find that fine-tuning on less visually appealing data can degrade the overall aesthetics; (ii) dataset size: the scale of these crawled multi-layer datasets is much smaller than that of conventional RGB image datasets. Consequently, fine-tuning on such datasets becomes less effective as the foundational generative models become increasingly powerful.

This paper leverages off-the-shelf powerful diffusion models to generate high-quality multi-layer transparent images, thereby bypassing the need for fine-tuning on specific datasets. To achieve this goal, this paper makes three key contributions: (i) LayerFLUX: We propose a training-free, single-layer transparent image generation system that utilizes a generate-then-matting scheme. Specifically, our approach leverages diffusion models to generate images with solid-colored backgrounds and uses a state-of-the-art image matting model to extract high-quality alpha masks for salient objects. We have named this system LayerFLUX, as it builds upon the latest diffusion transformer model, FLUX.1-[dev]. (ii) MultiLayerFLUX: We introduce a layout-then-layer scheme that composes multiple high-quality transparent layers generated by LayerFLUX according to a given layout, which can be obtained either from a reference image or generated using an LLM. This modular approach enables precise control over spatial composition while preserving the visual quality and alpha matte of each layer, resulting in our MultiLayerFLUX system. (iii) Transparent Image Preference Scoring Model: We develop a dedicated preference scoring model to assess the visual aesthetics of the generated transparent images. Figure 1 shows the high-quality synthetic multi-layer transparent images generated using MultiLayerFLUX.

To validate our designs, we first compare LayerFLUX with prior transparent image generation methods such as LayerDiffuse Zhang & Agrawala (2024). As shown in Figure 13, user studies on LAYER-BENCH (covering natural objects, sticker/text, and creative layers) confirm clear advantages. Next, we use MultiLayerFLUX to build PRISMLAYERS, a ∼200K multi-layer transparent dataset. After filtering, we obtain a 20K high-quality subset (PRISMLAYERSPRO), a 100K loosely filtered set (PRISMLAYERSPLUS), and a 1K photorealistic set (PRISMLAYERSREAL). Fine-tuning ART Pu et al. (2025) on PRISMLAYERSPRO yields ART+, which user studies (Figure 2) prefer in 57–60% of cases for prompt alignment, harmonization, and layer quality. Empirically, ART+ even approaches the quality of holistic single-layer images from FLUX.1-[dev]. These results highlight the critical role of high-quality multi-layer datasets in advancing next-generation transparent image generation, and we expect our open-source dataset to provide a strong foundation for future work.

## 2 RELATED WORK

**Transparent layer generation.** The task of transparent image generation has garnered increasing attention due to its high demand in interactive content creation, such as graphic design and presentation slides. Existing efforts can be broadly classified into two categories: (i) *Single-layer transparent image generation*: The representative LayerDiffuse Zhang & Agrawala (2024) introduces both a transparent layer generation model and a background-conditioned variant that produces layers consistent with the background. Text2Layer Zhang et al. (2023) directly trains a text-to-two-layer generation model that converts a text prompt into a foreground layer with a high-quality alpha matte and a corresponding coherent background layer. LayeringDiff Kang et al. (2025) proposes a two-stage approach: first, it generates a composite image using an off-the-shelf generative model, and then it disassembles this image into its foreground and background components via a segmentation or matting model, augmented by a diffusion-based layer separation model that refines the foreground and inpaints the background. (ii) *Multi-layer transparent image generation*: In contrast, LayerDiff Huang et al. (2024) introduces a layer-collaborative diffusion model capable of generating up to four layers from a text prompt. Following this direction, PSDiffusion Huang et al. (2025a) and DreamLayer Huang et al. (2025b) further explore harmonized generation mechanisms, utilizing global layout alignment and attention-based interaction to ensure coherence across multiple layers. ART Pu et al. (2025) employs an anonymous region transformer for variable multi-layer transparent image generation. Unlike the most related MULAN Tudosiu et al. (2024), which uses a top-down approach to decompose a monocular RGB image into a stack of RGBA layers representing the background and isolated instances, our approach adopts a bottom-up scheme. We first generate high-quality transparent layers and subsequently compose them into a unified graphic design. We also show that the visual aesthetic score statistics of the constructed PRISMLAYERS significantly outperform those of previous datasets.

**Graphic design generation.** Recently, an increasing number of studies Lin et al. (2024b); Weng et al. (2024); Lin et al. (2024a); Shabani et al. (2024); Kikuchi et al. (2024); Lin et al. (2023); Jia et al. (2023); Inoue et al. (2024); Peng et al. (2025) have shifted their focus from conventional photorealistic image generation to graphic design generation, driven by its substantial business value. One common approach involves generating all image layers simultaneously. For example, COLE Jia et al. (2023) and OpenCOLE Inoue et al. (2024) begin with a brief user prompt and employ multiple LLMs and diffusion models to iteratively generate each element of the final image. In contrast, Graphist Cheng et al. (2024) proposes a hierarchical layout generation system that creates graphic compositions from an unordered set of design elements. In this paper, we focus on building open, high-quality multi-layer transparent image datasets to facilitate future work on closing the gap between multi-layer generation and conventional single-layer text-to-image models. We also discuss the connections between our benchmark and previous multi-layer transparent image generation datasets in Table 1.

## 3 PRISMLAYERS: A HIGH-QUALITY MULTI-LAYER TRANSPARENT IMAGE DATASET

We introduce PRISMLAYERS, a synthetic dataset of ∼200K multi-layer transparent images, each with a global caption, layer-wise captions, RGB layers, and precise alpha mattes. All samples are filtered via our Transparent Image Preference Score (TIPS) model (Sec. 3.4) and Artifact Classifier (Sec. 3.2). From this, we curate a high-quality subset of 20K (PRISMLAYERSPRO) and a broader 100K set (PRISMLAYERSPLUS) by automatic filtering. Additionally, we construct a 1K photorealistic dataset, PRISMLAYERSREAL. We then present dataset statistics and the curation pipeline, followed by our key technical contributions: LayerFLUX and MultiLayerFLUX (Sec. 3.3).

### 3.1 PRISMLAYERS STATISTICS

**Statistics on the number of layers.** We analyze the distribution of transparent layer counts in PRISMLAYERS and PRISMLAYERSPRO. For PRISMLAYERS, each image contains an average of 7 layers (median: 6), with 79% of samples containing above 3 layers, while PRISMLAYERSPRO has an average of 9 layers (median: 9), with 90% of samples containing above 3 layers. This indicates that PRISMLAYERS effectively captures a wide range of visual complexity. Figure 3 (a) provides a more detailed illustration of the transparent layer count distribution.

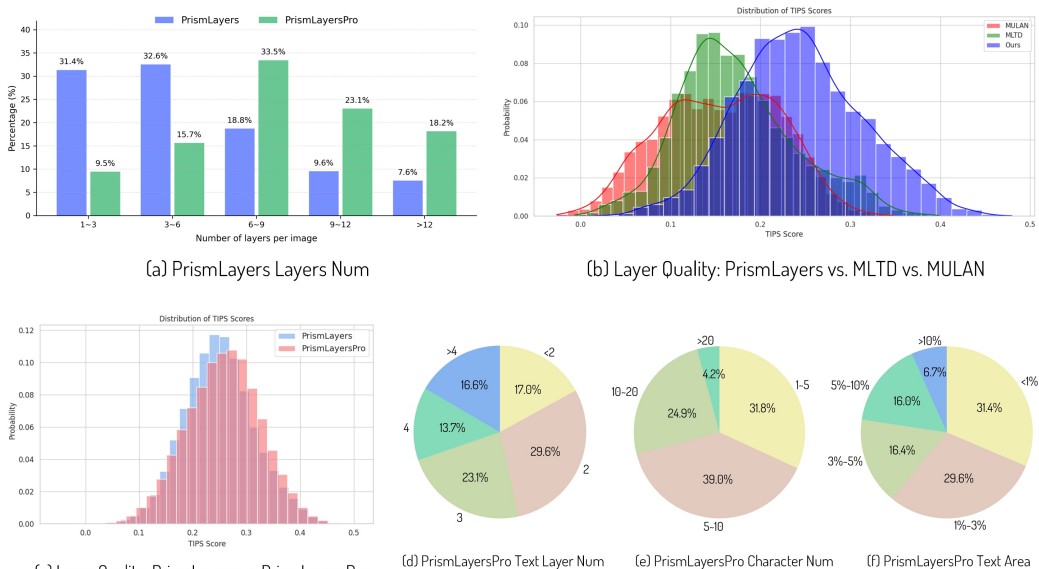

Figure 3: Illustrating the key dataset statistics on PRISMLAYERS and PRISMLAYERSPRO. Subfigure (a) shows the layer number distribution for PRISMLAYERS and PRISMLAYERSPRO Subfigures (b) and (c) compare the Layer Quality (TIPS scores) of the PRISMLAYERS dataset with the MLTD and MULAN datasets and PRISMLAYERS with PRISMLAYERSPRO Subfigures (d) through (f) detail the distribution of text layer count, character count, and text area within PRISMLAYERSPRO.

**Statistics on the aesthetics of layers.** A key contribution of this open-source dataset is the provision of aesthetically pleasing transparent layers, addressing the limited visual quality found in existing multi-layer datasets. As shown in Figure 3 (b), quantitative evaluations using our Transparent Image Aesthetic Scoring (TIPS) model illustrate the aesthetic distributions of PRISMLAYERS, MULAN Tudosiu et al. (2024), and MLTD Pu et al. (2025). Figure 3 (c) further compares the aesthetic quality between PRISMLAYERS and PRISMLAYERSPRO, PRISMLAYERSPRO increases the layers per image while achieving higher aesthetic quality compared to PRISMLAYERS. Figure 4 visualizes qualitative comparisons between PRISMLAYERS and PRISMLAYERSPRO. Our results show that PRISMLAYERS consistently provides higher-quality layers, with the open-source subset PRISMLAYERSPRO achieving the best overall aesthetic quality.

**Statistics of visual text layers.** High-quality visual text rendering is essential for multi-layer transparent image generation, as textual elements play a central role in many business-centric visual designs Liu et al. (2024b;c). PRISMLAYERS contains a large number of accurately rendered text layers, each isolated in a separate transparent channel. Figure 3 (d), (e), and (f) present statistics on the number of text layers per image, the number of characters per instance, and the area ratio of text layers.

**Statistics of different visual styles.** In the middle of Figure 1, we illustrate the distribution of transparent layers across different styles in PRISMLAYERSPRO, which contains 21 distinct styles. The top five most frequent styles are 'toy', 'melting silver', 'line draw', 'ink', and 'doodle art'

**Comparison with existing transparent datasets.** Table 1 presents a comparison with previously existing multi-layer transparent image datasets. We position PRISMLAYERSPRO as the first open, high-quality synthetic dataset that supports a diverse range of layers, high-quality alpha mattes, and excellent aesthetic quality. We believe PRISMLAYERSPRO can serve as a solid foundation for future efforts in building better multi-layer transparent image generation models.

### 3.2 PRISMLAYERS DATASET CURATION PROCESS

**Multi-layer prompts and semantic layout from crawled data.** Ⓐ → ❶ → Ⓑ We begin by collecting an internal dataset of 800K multi-layer graphic designs sourced from various commercial websites. Each design instance consists of multiple transparent layers, including background elements,

| Dataset | # Samples | # Layers | Open Source | Source Data | Alpha Quality | Aesthetic |
|---|---|---|---|---|---|---|
| Multi-layer Dataset Zhang & Agrawala (2024) | $\sim 1$ M | 2 | ✗ | commercial, generated | good | good |
| LAION-L$^2$I Zhang et al. (2023) | $\sim 57$ M | 2 | ✗ | LAION | normal | normal |
| MLCID Huang et al. (2024) | $\sim 2$ M | [2,3,4] | ✗ | LAION | poor | poor |
| MLTD Pu et al. (2025) | $\sim 1$ M | $2 \sim 50$ | ✗ | Graphic design website | good | normal |
| MAGICK Burgert et al. (2024) | $\sim 150$ K | 1 | ✓ | Synthetic | good | good |
| MuLAn Tudosiu et al. (2024) | $\sim 44$ K | $2 \sim 6$ | ✓ | COCO, LAION | poor | poor |
| Crello Yamaguchi (2021) | $\sim 20$ K | $2 \sim 50$ | ✓ | Graphic design website | normal | poor |
| PRISMLAYERS | $\sim 200$ K | $2 \sim 50$ | ✓ | Synthetic | good | good |
| PRISMLAYERSPLUS | $\sim 100$ K | $2 \sim 50$ | ✓ | Synthetic | good | good+ |
| PRISMLAYERSPRO | $\sim 20$ K | $2 \sim 50$ | ✓ | Synthetic | good | excellent |
| PRISMLAYERSREAL | $\sim 1$ K | $1 \sim 8$ | ✓ | Synthetic | good | excellent |

Table 1: Comparison with previous multi-layer transparent image datasets. This table presents a detailed comparison of publicly available and newly proposed multi-layer image datasets across several key dimensions, including the number of samples, the maximum number of layers, open-source status, source data origin, and two subjective quality metrics: Alpha Quality and Aesthetic appeal.

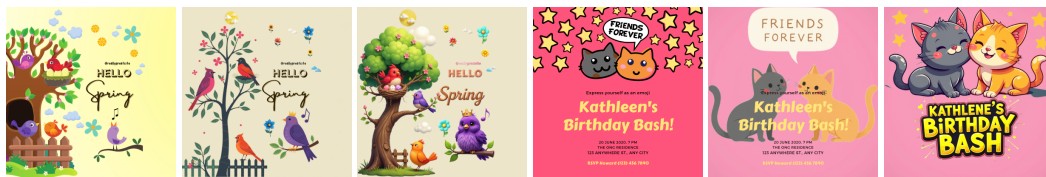

Figure 4: Illustrating the aesthetic quality of the crawled data (columns 1 and 4), synthetic data (columns 2 and 5), and high-quality synthetic data generated with a style prompt (columns 3 and 6).

decorations, text, and icons. To enrich the semantic understanding of each instance, we employ an off-the-shelf LLM—Llava 1.6 Liu et al. (2024a)—to generate captions for both individual transparent layers and the fully composed images. This process yields annotations comprising 800K multi-layer prompts and their corresponding semantic layouts, effectively capturing both the visual composition and the intended design semantics. We also extract the original metadata specifying the layer ordering for each graphic. For the filtered PRISMLAYERSPLUS and PRISMLAYERSPRO, we further enhance semantic richness by using GPT-4o to generate high-quality layer-wise captions.

**Synthetic multi-layer transparent images with MultiLayerFLUX.** **B** → **②** → **C** With the constructed 800K multi-layer prompts and corresponding semantic layout information, we apply a novel model, MultiLayerFLUX, to transform the layer-wise prompts into multiple transparent layers, each generated separately using a single-layer transparent image generation engine such as LayerFLUX, as illustrated in Sec. 3.3. We then composite these transparent layers onto a shared canvas, preserving the correct stacking order and ensuring seamless integration across layers. In total, creating the entire 800K multi-layer images takes around 7,000 A100 GPU hours.

**Artifact multi-layer transparent image filter.** **C** → **③** → **D** As MultiLayerFLUX generates each transparent layer separately and then combines them following the layer order, we observe severe artifacts in some synthetic multi-layer images. These artifacts include duplicate or similar layers positioned in conflicting spatial arrangements or exhibiting substantial and unreasonable overlap, as shown in Figure 6. To address this issue, we construct a reliable artifact classifier to further filter out flawed multi-layer transparent images. We begin by manually annotating severe artifacts in a subset of 8K synthetic multi-layer images with high aesthetic scores. Then, we train an artifact classifier by fine-tuning BLIP-2 Li et al. (2023) to predict confidence scores indicating whether a composed multi-layer transparent image contains such artifacts—e.g., conflicting layer placements or unreasonable overlap. To ensure the quality of the final dataset, we apply the trained classifier to select a subset of 200K synthetic multi-layer transparent images, forming PRISMLAYERS.

**High-quality reference layout pool.** **D** → **E** The aforementioned Artifact Classifier performs image-level structural assessment. Next, we perform visual quality filtering using an aesthetic predictor aes. We rank images with different numbers of layers based on their aesthetic scores, then select a fixed proportion of the highest-scoring images from each group to form an 80K-image high-quality reference layout pool.

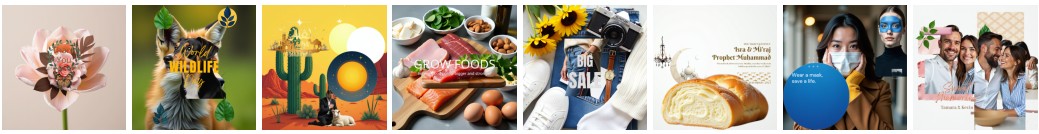

Figure 5: **Dataset Curation Pipeline of PRISMLAYERS, PRISMLAYERSPLUS, and PRISMLAYERSPRO.** We first extract semantic layouts from a database of 800K crawled multi-layer graphic design images. Then, we apply MultiLayerFLUX to generate high-quality multi-layer transparent images. An Artifact Classifier is used to evaluate the quality of each composed image, discarding low-quality results to construct PRISMLAYERS. We also apply the Transparent Image Preference Score (TIPS) model to assess the quality of individual transparent layers. By filtering for aesthetic quality and balancing the number of layers, we collect an 80K-image reference layout pool. We sample layouts from this pool and regenerate them with style prompts, followed by quality evaluation and manual selection, forming our high-quality multi-layer dataset, PRISMLAYERSPLUS and PRISMLAYERSPRO.

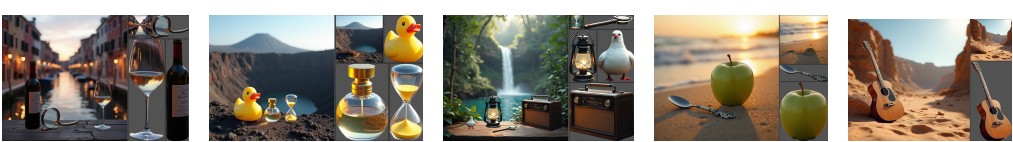

Figure 6: Illustrating the artifact multi-layer transparent images that our classifier can identify and filter out.

**Layer-wise quality filter, styled prompt rewrite, and human selection.** Ⓔ → ❺ → ❸ + ❹ (+❻) → Ⓕ (Ⓖ) To improve layer quality, we construct two refined subsets from PRISMLAYERS: 20K PRISMLAYERSPRO and 100K PRISMLAYERSPLUS. The pipeline involves three steps: 1) Styled prompt rewrite: we define 21 style keywords and sample layouts from the 80K reference pool. For each style, 2K layouts (for PRISMLAYERSPRO) and 8K (for PRISMLAYERSPLUS) are selected. Their layers are pasted onto a gray background and fed to GPT-4o, which rewrites captions with style directives. MultiLayerFLUX then regenerates transparent layers from these captions. 2) Quality Evaluation: we train the TIPS model on a collection of transparent images from our PRISMLAYERS, single-layer images generated by LayerDiffuse Zhang & Agrawala (2024), and our reproduction of LayerDiffuse based on FLUX.1-[dev]. The TIPS model is combine with Artifact Classifier to evaluate both whole images and layers. 3) Human selection (for PRISMLAYERSPRO): top-quality samples are manually selected by removing obvious failures with reference to the scores of quality evaluation, while without the human selection, automatically filtered samples form PRISMLAYERSPLUS. In practice, generate rate 20K PRISMLAYERSPRO with GPT refined annotations takes around 480 A100 GPU hours, while generating 100K PRISMLAYERSPLUS takes around 2,400 A100 GPU hours.

**Photorealistic multi-layer image synthesis.** Our approach primarily focuses on the design-oriented synthesis of multi-layer datasets. Nevertheless, as shown in Figure 7, we also explore photorealistic multi-layer image synthesis leveraging the prior knowledge from our collected multi-layer graphic design images, resulting in a small but high-quality 1K dataset, PRISMLAYERSREAL, and we discuss more details in Appendix.N.

Figure 7: Illustrating the samples in PRISMLAYERSREAL.

**Discussion.** A natural question is whether the results exhibit cross-layer coherence. We acknowledge this limitation, as synthetic multi-layer images cannot fully ensure consistency, though partial mitigation is achieved via human selection. Importantly, we observe that training the recent ART

Figure 9: Attention maps between the *suffix text token* and *visual tokens*. We observe a clearly higher attention response in the background area with accurate boundary patterns.

model Pu et al. (2025) on our filtered high-quality dataset, yields noticeably improved coherence, underscoring the importance of high-quality supervision.

## 3.3 LayerFLUX and MultiLayerFLUX

In this section, we present the mathematical formulation of the multi-layer transparent image generation task, followed by key insights and implementation details of our LayerFLUX and MultiLayerFLUX.

**Formulation.** The transparent image generation task aims to train a generative model that transform the input global text prompt $\mathbf{T}_{\text{global}}$ and the optional regional text prompts $\{\mathbf{T}_{\text{region}}^i\}_{i=1}^N$ into an output consisting of a set of transparent layers $\{\mathbf{I}_{\text{RGBA}}^i\}_{i=1}^N$ that can form a high-quality multi-layer image $\mathbf{I}_{\text{global}}$, and each layer is with accurate alpha channels $\{\mathbf{I}_{\text{alpha}}^i\}_{i=1}^N$. This task degrades to a single-layer transparent image generation task when $N = 1$. Following the latest ART Pu et al. (2025),

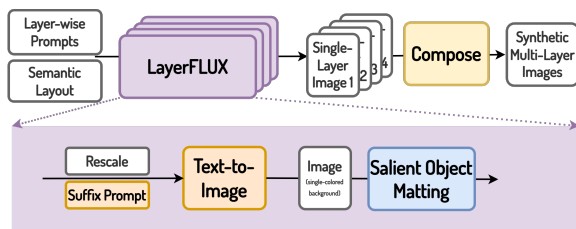

Figure 8: LayerFLUX and MultiLayerFLUX Framework.

we apply a flow matching model to model the multi-layer transparent image generation task by performing the latent denoising on the concatenation of both the global visual tokens and the regional visual tokens.

**LayerFLUX.** As shown in Figure 8, we build the LayerFLUX with two key designs, including the suffix prompt scheme and the additional salient object matting to predict the accurate alpha mattes.

Inspired by MAGICK Burgert et al. (2024), we design a series of tailored suffix prompts to guide diffusion models in generating images with single-colored, uniform backgrounds. These controlled conditions ensure that the foreground elements are clearly delineated, thereby simplifying the isolation process. Our implementation involves simply appending the suffix prompt "*isolated on a gray background*" to the original text prompt. Figure 9 visualizes the attention maps between the suffix tokens and the visual tokens. We also compare the results of using alternative suffix prompts by replacing the word "*gray*" with other colors. A detailed analysis of different suffix prompt effects is provided in Appendix.L,M.

To extract accurate alpha mattes, we explore and evaluate multiple state-of-the-art image matting techniques, including SAM2 Ravi et al. (2024), BiRefNet Zheng et al. (2024), and RMBG-2.0 RMB, to seperate the foreground from the background. This step is critical for producing high-quality, transparent images that can be seamlessly integrated into multi-layer compositions. We empirically find that RMBG-2.0 achieves the best matting quality, and we choose it as our default method.

**MultiLayerFLUX.** We construct the MultiLayerFLUX framework by stacking the outputs from the above-mentioned LayerFLUX according to the given layer-wise prompts and semantic layout. We observe that simply applying LayerFLUX to generate each layer within a fixed square canvas tends to produce objects with an unnatural square shape. Instead of generating each layer in a fixed square canvas, we preserve the original aspect ratio of each transparent layer and use FLUX.1-[dev] to generate images at varying resolutions, fixing the longer side to 1024. Each generated transparent layer is then resized to fit the corresponding bounding boxes based on the semantic layout information, and the layers are composited according to the layer-order annotations, resulting in the final synthetic multi-layer transparent images.

### 3.4 Transparent Image Quality Assessment

Existing image quality assessment models Kirstain et al. (2024); Wu et al. (2023); Xu et al. (2024) are primarily trained to predict human preferences for conventional RGB images, and thus are not well suited for evaluating transparent images with alpha mattes. To address this gap, we propose a dedicated quality scoring model tailored for transparent layer images. The core idea is to distill ensembled preference signals—aggregated from multiple RGB-oriented models—into a model specialized for transparent image quality, thereby mitigating model-specific biases.

**Transparent image preference dataset.** We first collect a transparent image preference (TIP) dataset of more than 100K win-lose pairs by gathering three types of data resources, including those generated with LayerFLUX and LayerDiffuse. We use multiple image quality scoring models to rate the quality of each transparent layer, including Aesthetic Predictor V2.5 aes, Image Reward Xu et al. (2024), LAION Aesthetic Predictor lai, HPSV2 Wu et al. (2023), and VQA Score Lin et al. (2024c). Then, we compare each pair of transparent layers based on the weighted sum of the scores predicted by the aforementioned quality scoring models. Here, we assume that the alpha mask quality of most transparent layers generated with our LayerFLUX and LayerDiffuse methods is satisfactory.

**Transparent image preference score.** We train the transparent image preference scoring model by fine-tuning CLIP on the TIP dataset. For each pair of transparent images with preference labels, we choose loss function $\mathcal{L}_{\text{pref}} = (\log 1 - \log \mathbf{p}_w)$, where $\mathbf{p}_w$ is the probability of the win image being the preferred one, and we compute the $\mathbf{p}_w$ as:

$$\mathbf{p}_w = \frac{\exp\left(\tau \cdot f_{\text{CLIP-V}}(\mathbf{I}^w) \cdot f_{\text{CLIP-T}}(\mathbf{T})\right)}{\exp\left(\tau \cdot f_{\text{CLIP-V}}(\mathbf{I}^w) \cdot f_{\text{CLIP-T}}(\mathbf{T})\right) + \exp\left(\tau \cdot f_{\text{CLIP-V}}(\mathbf{I}^l) \cdot f_{\text{CLIP-T}}(\mathbf{T})\right)}, \tag{1}$$

where $f_{\text{CLIP-V}}(\cdot)$ and $f_{\text{CLIP-T}}(\cdot)$ represent the CLIP visual encoder and text encoder separately. $\mathbf{I}^w$ and $\mathbf{I}^l$ represent the prefered and disprefered transparent image.

During the evaluation, we compute the transparent image preference score as follows:

$$\mathbf{p} = f_{\text{CLIP-V}}(\mathbf{I}) \cdot f_{\text{CLIP-T}}(\mathbf{T}), \tag{2}$$

where we directly use the dot product between the normalized CLIP visual embedding and the CLIP text embedding as the transparent image preference score, abbreviated as TIPS for convenience.

## 4 Experiment

### 4.1 Setting

We conduct all experiments with the latest FLUX.1 [dev] flu model. Our multilayer experiments on PrismLayers are based on fine-tuning the previous SOTA multilayer transparent image generation method, ART Pu et al. (2025). The new model is named ART+. For fine-tuning detail, we use 20K training iterations, a global batch size of 4, an image resolution of 512×512, and a learning rate of 1.0 with the Prodigy optimizer, followed by fine-tuning at a larger resolution of 1024×1024 with 10K training iterations.

| Method | MLTD 800K | PrismLayers | | |
|--------|-----------|-------------|----------|------------|
| | | 200K | PRO(20K) | PLUS(100K) |
| ART | ✓ | | | |
| ART+(20k scratch) | | ✓ | ✓ | |
| ART+(20k) | ✓ | | ✓ | |
| ART+(100k) | ✓ | | | ✓ |

Table 2: Training data configurations for ART and ART+ models. The data in the first two columns are utilized for the initial step of FLUX.1 [dev] fine-tuning, and the data in the last two columns are used for subsequent high-quality fine-tuning.

Instead of assessing the model's performance solely on the Design-Multi-Layer-Bench Pu et al. (2025)—a benchmark consisting of crawled multilayer graphic designs, most of which follow a similar flat style—we propose evaluating it on a more diverse and creative set we call FLUX-Multi-Layer-Bench. This benchmark is chosen to quantify the gap between generated multi-layer graphic designs and the holistic single-layer image designs produced by the latest text-to-image generation models. A more detailed introduction to both benchmarks is provided in Appendix.D.

### 4.2 ART+: Improving ART with PrismLayers

To ensure a fair comparison, we trained multiple ART+ models, as detailed in Table 3, which will be elaborated upon in this section.

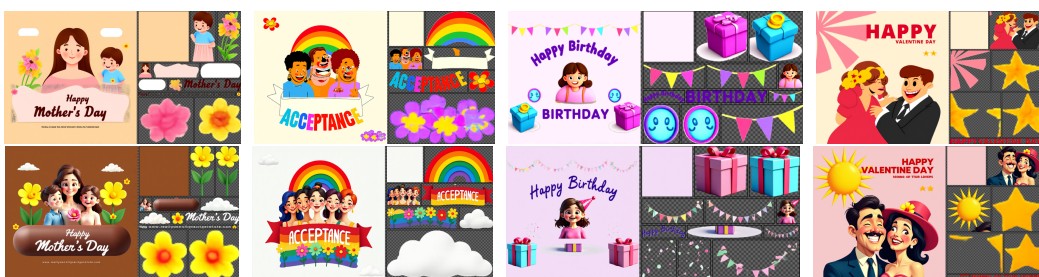

Figure 10: Qualitative comparison results between ART (top row) and ART+ (bottom row).

| Method | Design-Multi-Layer-Bench | | | | FLUX-Multi-Layer-Bench | | | |
|---|---|---|---|---|---|---|---|---|
| | $FID_{merged}$ ↓ | TIPS | PSNR | SSIM | $FID_{merged}$ ↓ | TIPS | PSNR | SSIM |
| ART Pu et al. (2025) | 18.34 | 16.84 | 27.41 | 0.9490 | 30.04 | 16.64 | 26.99 | 0.9502 |
| MultiLayerFLUX | 21.29 | 19.90 | - | - | 29.64 | 20.65 | - | - |
| ART+(20k scratch) | 26.53 | 18.91 | 28.12 | 0.9544 | 26.07 | 19.42 | 28.12 | 0.9559 |
| ART+(20k) | 21.66 | 18.82 | 27.90 | 0.9536 | 25.23 | 18.98 | 28.06 | 0.9560 |
| ART+(100k) | 25.11 | 18.13 | 26.71 | 0.9423 | 25.63 | 18.27 | 26.80 | 0.9455 |

Table 3: Quantitative comparison of our ART+ with the state-of-the-art ART and our proposed MultiLayerFLUX. The best metric value in each column is indicated with an underline.

**User Study Evaluation.** To assess the effectiveness of our dataset and fine-tuning strategy, we conduct a user study comparing ART+(20k scratch) with the original ART, PrismLayers, and PrismLayersPro. The study involves 40 representative samples from FLUX-Multi-Layer-Bench, with over 20 participants evaluating three key dimensions: (i) *Layer Quality* (visual aesthetics and alpha fidelity), (ii) *Global Harmonization* (inter-layer coherence), and (iii) *Prompt Following* (alignment with input prompts).

As shown in Figure 2, ART+(20k scratch) outperforms the original ART with average win rates of 57.9% in layer quality and 59.3% in prompt following. It also surpasses MultiLayerFLUX in global harmonization (45.1% win rate), validating the impact of combining high-quality supervision with task-specific tuning.

**Quantitative Results.** Table 3 presents a comprehensive comparison between our ART+ model and the current state-of-the-art (SOTA) model on two benchmarks with different data distributions. For ART+, the MLTD (800K) and PrismLayers (200K) datasets are used for fine-tuning the base FLUX model (similar to the ART approach), while PrismLayersPro and PrismLayersPlus are used for subsequent high-quality fine-tuning following Dai et al. (2023). We evaluated four key metrics: $FID_{merged}$ and TIPS for image and layer quality, and PSNR and SSIM for transparent decoding quality. The results consistently demonstrate that our ART+ model achieves superior performance across these metrics, establishing it as the new SOTA.

For the $FID_{merged}$ score, our ART+ significantly outperforms ART on FLUX-Multi-Layer-Bench. We also provide additional qualitative comparison results to support this finding. An interesting phenomenon, however, is that ART demonstrates superior performance on Design-Multi-Layer-Bench. This can be attributed to the significant similarity between ART's training data distribution and this benchmark, a more detailed analysis of which is provided in Appendix.D.

Regarding the TIPS scores, we note a partial data overlap between the training data of ART+(20k scratch) and the TIPS evaluation set. While this might be a concern for fairness, our more extensive testing of ART+(20k) and ART+(100k) — where this overlap does not exist — effectively mitigates this issue. All results consistently show that the layer quality of ART+ is far superior to ART and approaches that of the original FLUX.

Across all metrics, the performance of ART+(100k) is comparable to ART+(20k) and surpasses ART. This demonstrates that our automated approach can efficiently build high-quality, reproducible training data on a large scale without the need for human intervention.

**Qualitative MultiLayer Results.** Figure 11 presents qualitative results comparing our MultiLayerFLUX with the fine-tuned ART+, while Figure 10 shows qualitative comparisons between ART and the fine-tuned ART+. We observe that ART+ achieves significantly better global harmonization than

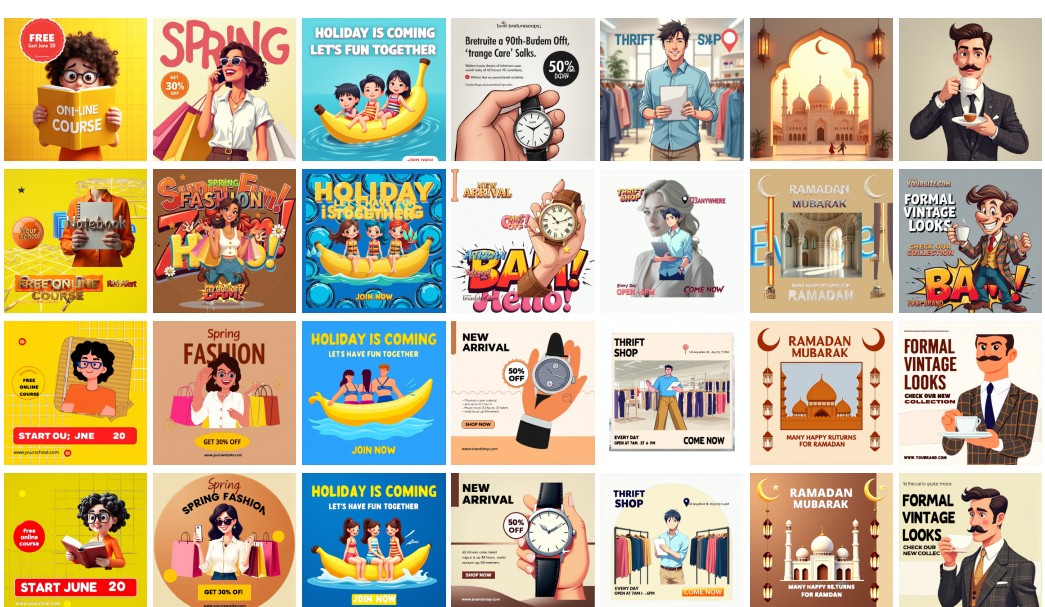

Figure 11: Qualitative comparison results between MultiLayerFLUX (top row) and ART+ (bottom row).

Figure 12: Qualitative comparison results between FLUX.1-[dev] (1st row), MultiLayerFLUX (2nd row), ART (3rd row), and ART+ (4th row) across 7 cases (columns). The rightmost columns show composed multi-layer images.

MultiLayerFLUX and better layer quality than ART, separately. These comparisons reveal that the fine-tuned ART+ achieves an excellent balance between layer quality and global harmonization.

**Comparison to FLUX.** Figure 12 compares the merged multi-layer image generation results with the reference ideal images generated directly with FLUX.1-[dev]. We can see that our ART+ significantly outperforms ART and MultiLayerFLUX, achieving aesthetics very close to those of the original modern text-to-image generation models.

## 5 CONCLUSION

This paper has tackled the significant gap in multi-layer transparent image generation by assembling and releasing four open, ultra-high-fidelity datasets—PRISMLAYERS (200K samples), PRISMLAYERSPLUS (100K samples), PRISMLAYERSPRO (20K samples), and PRISMLAYERSREAL (1K samples)—each annotated with precise alpha mattes. To produce this data on demand, we devised a training-free synthesis pipeline that harnesses off-the-shelf diffusion models, and we built two complementary methods: LayerFLUX and MultiLayerFLUX. After rigorous artifact filtering and human validation, we fine-tuned the ART model on PRISMLAYERSPRO to obtain ART+, which outperforms the original ART in 60% of head-to-head user studies and matches the visual quality of top text-to-image models. By establishing this open dataset, synthesis pipeline, and strong baseline, we lay a solid foundation for future research and applications in precise, editable, and visually compelling multi-layer transparent image generation.

## ETHICS STATEMENT

Our work does not appear to raise major ethical concerns. The dataset used in this work is entirely synthetic, generated without involving real human subjects. During the data generation process, we applied filtering mechanisms to exclude outputs that may pose potential ethical concerns.

## REPRODUCIBILITY STATEMENT

All experiments and synthetic data generation were conducted on NVIDIA A100 GPUs. To facilitate reproducibility, we provide a detailed description of the pipeline and algorithms used to construct the dataset, which are straightforward and reproducible. We also illustrate all experimental settings, training details, and evaluation protocols within the paper and appendix. In addition, we will release the dataset in the future to ensure reproduction and enable further research.

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

**LLM usage.** We use Large Language Models (LLMs) to assist in synthetic image annotation and text-to-image (T2I) prompt generation.

**A. Details of Suffix Prompt Templates** Table 4 illustrates the detailed suffix prompt templates we adopted for LayerFLUX.

| Method | detailed prompt |
|--------|-----------------|
| SuffixPrompt A | on a solid plain gray background. |
| SuffixPrompt B | with a clear, solid gray background. |
| SuffixPrompt C | on a solid single gray background. |
| SuffixPrompt D | floating with a background that is solid gray. |
| SuffixPrompt E | cut-out on a solid gray background. |
| SuffixPrompt F | standing on a background that is fully solid gray |
| SuffixPrompt G | without any surrounding details |
| SuffixPrompt H | isolated on a solid gray background |

Table 4: Effect of choosing different suffix prompt templates.

**B. Generating Multi-Page and Multi-Layer Transparent Slides.** We plan to extend our approach to generate multi-page, multi-layer transparent slides. Our framework not only produces single-layer transparent images but also assembles them into coherent slide decks with multiple pages. Each slide is constructed from several transparent layers, with each layer corresponding to different design elements. This modular, bottom-up strategy enables precise control over both the spatial layout and stylistic attributes of each slide, ensuring consistency across pages while preserving the flexibility to customize individual layers.

**C.Explanation of the Benchmarks** Here, we provide a detailed explanation of the composition of our two benchmarks: DESIGN-MULTI-LAYER-BENCH and FLUX-MULTI-LAYER-BENCH.

DESIGN-MULTI-LAYER-BENCH is a validation set originally introduced in the ART Pu et al. (2025). This dataset comprises 5,000 samples constructed from templates on popular graphic design platforms such as VistaCreate and Canva. A key characteristic of this benchmark is that its ground-truth reference images were manually created by human designers, though their style is primarily restricted to flat design. The data distribution of this benchmark is nearly identical to that of the ART training data.

In contrast, FLUX-MULTI-LAYER-BENCH is our newly developed validation set, also containing 5,000 samples. The core difference lies in its ground-truth reference images, which are high-quality, single-layer images generated by the FLUX-1-[dev] model. This benchmark is specifically designed to more accurately measure the visual fidelity gap between generated multi-layer images and state-of-the-art text-to-image models. Furthermore, this dataset encompasses a more diverse range of graphic design styles, thereby addressing the stylistic uniformity limitation of the former benchmark.

**D. Discussion about FID on the DESIGN-MULTI-LAYER-BENCH** Despite the significant improvements made by the ART+ model, its FID (Fréchet Inception Distance) score on the DESIGN-MULTI-LAYER-BENCH is unexpectedly higher than that of the original ART model. This counterintuitive result is primarily due to two core reasons:

First, the FID metric measures the statistical distance between feature distributions in the Inception-v3 feature space. The ground-truth reference images in the DESIGN-MULTI-LAYER-BENCH are heavily concentrated on a specific flat graphic design style. In contrast, our ART+ model was fine-tuned on the PrismLayers/PrismLayersPro dataset, which is designed to generate more diverse, higher-quality transparent layers across a broader range of styles. Consequently, the feature distribution of the ART+ model's outputs diverges significantly from the narrow, flat-design distribution of the DESIGN-MULTI-LAYER-BENCH, leading to the FID metric unfairly penalizing the model's increased diversity.

To more reliably assess the visual quality of the generated transparent layers, we constructed the FLUX-MULTI-LAYER-BENCH, whose ground-truth reference images are high-quality outputs from the state-of-the-art text-to-image model FLUX-1-[dev]. On this benchmark, our ART+ model achieves a lower FID score, which strongly validates the benefits of our training with PrismLayers. Additionally,

we also demonstrate that ART+ can generate multi-layer images with much better aesthetics, as shown in Figure 10 and 12.

In conclusion, FID primarily measures the closeness of two feature distributions, not necessarily the perceptual quality of the images themselves. Therefore, a fair comparison requires a test set that aligns with the model's target data distribution. The suboptimal performance of ART+ on the DESIGN-MULTI-LAYER-BENCH is a direct consequence of a data distribution mismatch. For future work, we welcome and encourage further exploration into developing a more suitable evaluation metric than FID.

**E. Side Effect of Suffix Prompt.** We admit that adding the suffix prompt is not a free lunch and report the results of adding the suffix prompt on the GenEval benchmark in Table 5. We can see that the prompt-following capability of the original text-to-image generation model slightly drops, while the visual aesthetics are maintained.

| Model | Overall | Single | Two | Counting | Colors | Position | Color |
|---|---|---|---|---|---|---|---|
| FLUX.1-[dev] | 0.657 | 0.978 | 0.816 | 0.716 | 0.801 | 0.228 | 0.405 |
| FLUX.1-[dev] + suffix prompt | 0.591 | 0.906 | 0.609 | 0.628 | 0.723 | 0.313 | 0.370 |

Table 5: Comparison results on GenEval.

**F. Technical Details of LayerDiffuse with FLUX.** Our implementation of Layerdiffuse with FLUX is built on FLUX.1-[Dev] with LoRA. Specifically, we convert the image in the MAGICK dataset to grayscale according to the alpha channel mask. After training, the model is capable of generating grayscale background images without the need for additional conditional inputs. Then, we train a transparency VAE decoder to enable the prediction of alpha channels. The decoder is trained on both the MAGICK dataset and an internal dataset, thereby enhancing its robustness and generalization. For the text sticker, we collect a 5k dataset and use GPT-4o to reception of the image.

**G. Experiment Results of LayerFLUX.** We construct a LAYER-BENCH to evaluate the quality of the single-layer transparent images generated by our LayerFLUX. The LAYER-BENCH consists of 1,500 prompts divided into three types of prompt sets: (i) one that primarily focuses on natural objects sampled from the MAGICK Burgert et al. (2024) set, where each prompt describes a photorealistic object; (ii) one centers on stickers and text stickers, where the text stickers contains visual text designed in creative typography and style to make the words stand out as part of the visual design; and (iii) one is about creative and stylistic objects. We construct the test set of stickers and text stickers by recaptioning sticker images crawled from the internet.

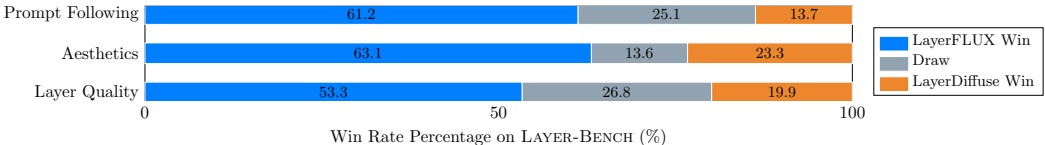

Figure 13: Illustrating the win-rate on single-layer transparent image generation benchmark LAYER-BENCH.

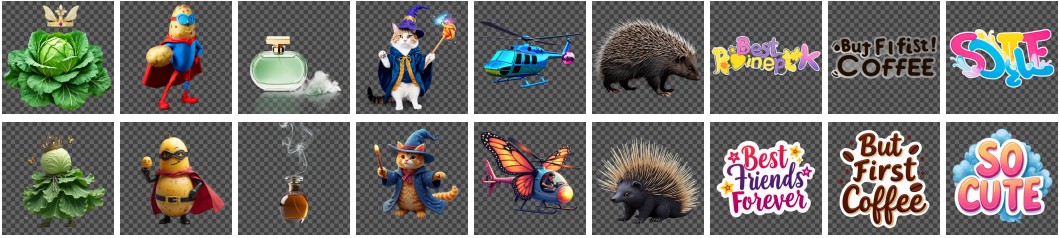

Figure 14: Qualitative comparison of results with SOTA on LAYER-BENCH. The first row shows the results generated with LayerDiffuse, while the second row shows the results generated with our LayerFLUX.

We compare our approach to the latest state-of-the-art transparent image generation LayerDiffuse Zhang & Agrawala (2024) by involving more than ∼ 20 participants from diverse backgrounds in AI, graphic

| # samples | TIPS (Layer Quality) | Composed Image Quality |
|---|---|---|
| Baseline (ART) | 0.114±0.077 | 4.674±0.373 |
| 10 | 0.110±0.076 | 4.684±0.543 |
| 100 | 0.130±0.086 | 4.938±0.418 |
| 1000 | 0.135±0.080 | 4.936±0.415 |

Table 6: Effect of the high-quality data scale.

| Method | Natural Object Layer Quality | | | Sticker Layer Quality | | | Creative Object Layer Quality | | |
|---|---|---|---|---|---|---|---|---|---|
| | HPSv2 ↑ | AE-V2.5 ↑ | TIPS ↑ | HPSv2 ↑ | AE-V2.5 ↑ | TIPS ↑ | HPSv2 ↑ | AE-V2.5 ↑ | TIPS ↑ |
| LayerDiffuse Zhang & Agrawala (2024) | 26.28 | 5.451 | 29.37 | 21.51 | 3.640 | 19.11 | 29.13 | 5.057 | 32.53 |
| LayerDiffuse w/ FLUX | 24.33 | 5.374 | 27.65 | 25.79 | 4.376 | 25.16 | 25.25 | 4.974 | 29.09 |
| Ours | 26.58 | 5.617 | 30.19 | 26.14 | 4.735 | 25.69 | 29.55 | 5.551 | 36.25 |

Table 7: Comparison with LayerDiffuse on LAYER-BENCH.

design, art, and marketing. We present system level comparison in Table 7 and the user study results and visual comparisons in Figure 13 and Figure 14. We can see that our LayerFLUX achieves better results across the three types of prompt sets, especially in the creative, stylistic, or text sticker prompt sets. For example, our LayerFLUX achieves better layer quality and prompt following than LayerDiffuse, with win-rates of 63.1% and 61.2% when evaluated on our LAYER-BENCH. One possible concern might be that LayerDiffuse is built on SDXL Podell et al. (2023) rather than FLUX.1-[dev]. We also fine-tune LayerDiffuse on existing transparent image datasets based on FLUX, but we find that the performance is even worse than that of the original LayerDiffuse based on SDXL. We infer that *a key reason is that the quality of data generated by these powerful models (like FLUX.1-[dev]) significantly outperforms that of existing transparent images available on the internet or predicted by existing models*. This widening quality gap makes it risky to fine-tune them directly. In summary, our training-free LayerFLUX can better maintain the original capabilities of the off-the-shelf text-to-image generation model, providing a solid foundation for a wide range of applications.

**H. Effect of salient object matting model choice.** How to extract high-quality alpha channels is critical for constructing high-quality single-layer transparent images. We study the influence of different salient object matting models, such as SAM2, BiRefNet, and RMBG-2.0, and summarize the comparison results on LAYER-BENCH in Table 8. We primarily consider the visual aesthetics of the transparent layers after matting and report the quantitative results. Additionally, we visualize the qualitative comparison results in Figure 15. We empirically find that RMBG-2.0 achieves the best results and adopt it as the default model.

**I. Prompt of the Creative Caption Generation** Compared to the common images in the MAGICK dataset, creative images reflect the model's ability to generate less frequent and more novel visual content. To evaluate this capability of our method, we constructed a test set consisting of 500 creative prompts generated by GPT-4o, ensuring diversity and originality in the evaluation dataset. We mainly focus on single objective description generation

**J. Prompt of Multi-layer Style-align Recaption Instruction** Given a reference layer of a multi-layer image, we leverage the visual recognition capabilities of GPT-4o and style-align reception instruction to transfer the original layer caption to a specific style caption. Specifically, we paste the original layer to the center of a gray background image while keeping the aspect ratio. Then, the style-specific instruction and the gray background layer image are fed to GPT-4o. Also, for the generation of ART, we use a similar instruction prompt to transfer the overall writing and style of the global caption.

**K. How to choose the suffix prompt?**

To understand how the suffix prompt helps the transparent layer generation task, we analyze the attention maps between the background regions and the color text tokens within the suffix prompt in Table 9, where we observe that the *"gray"* token achieves the best attention map response. We further conducted a series of experiments to compute $mIoU_{FG}$ and $mIoU_{BG}$ by calculating the mean IoU between the binary attention mask and the mask predicted by an image matting model to demonstrate the effect of choosing different suffix prompts quantitatively. In addition, we compute the mean square error between the attention map and the matting mask using $MSE_{BG}$ and $MSE_{FGLeak}$, where the latter metric reflects the degree of information leakage from the background to the foreground regions. We

| Method | Natural Object Layer Quality | | | Sticker Layer Quality | | | Creative Object Layer Quality | | |
|---|---|---|---|---|---|---|---|---|---|
| | HPSv2 ↑ | AE-V2.5 ↑ | TIPS ↑ | HPSv2 ↑ | AE-V2.5 ↑ | TIPS ↑ | HPSv2 ↑ | AE-V2.5 ↑ | TIPS ↑ |
| SAM2 | 26.24 | 5.374 | 30.03 | 26.04 | 4.556 | 24.49 | 30.01 | 5.251 | 36.76 |
| BiRefNet | 26.03 | 5.548 | 29.26 | 26.08 | 4.719 | 25.62 | 29.09 | 5.503 | 35.24 |
| RMBG-2.0 | 26.58 | 5.617 | 30.19 | 26.14 | 4.735 | 25.69 | 29.55 | 5.551 | 36.25 |

Table 8: Effect of choosing different salient object matting models.

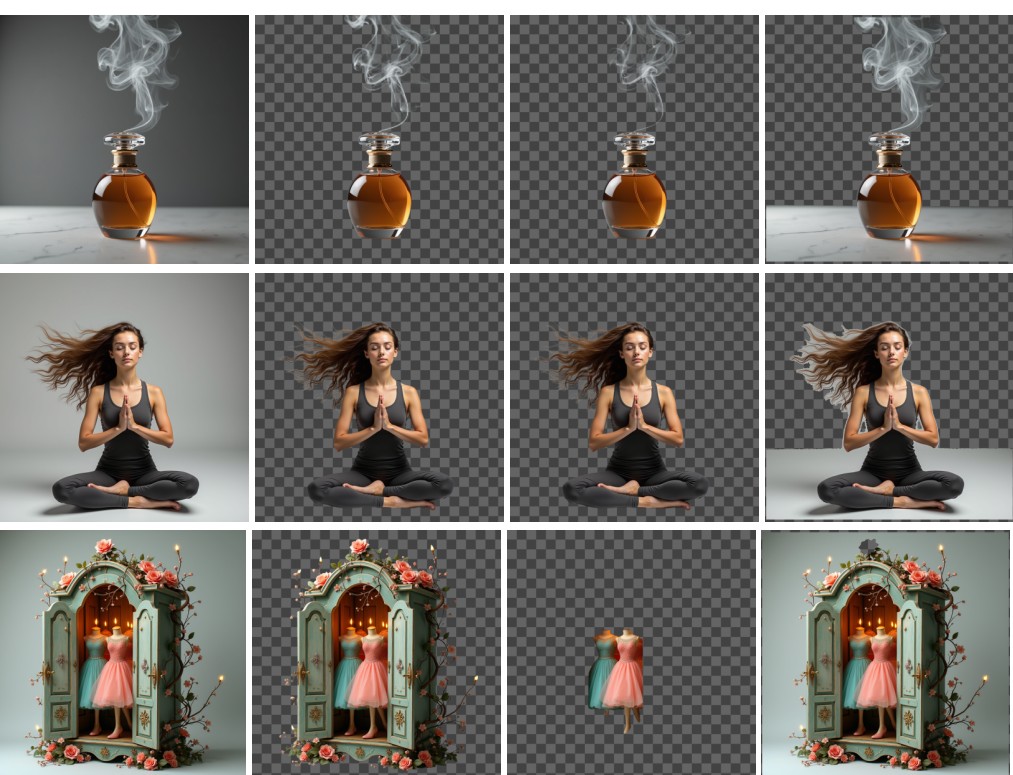

Figure 15: Qualitative comparison of different salient object matting models. From left to right, we show the matted results with RMBG-2.0, BiRefNet, and SAM2.

compute these metrics as follows:

$$\text{IoU}_{\text{BG}} = \frac{|(1 - \mathbf{M}) \cap \overline{\mathbf{A}}|}{|(1 - \mathbf{M}) \cup \overline{\mathbf{A}}|}, \qquad \text{MSE}_{\text{BG}} = \frac{1}{N} \sum_{i=1}^{N} ((1 - \mathbf{M}_i) - \mathbf{A}_i)^2, \tag{3}$$

$$\text{IoU}_{\text{FG}} = \frac{|\mathbf{M} \cap (1 - \overline{\mathbf{A}})|}{|\mathbf{M} \cup (1 - \overline{\mathbf{A}})|}, \qquad \text{MSE}_{\text{FGLeak}} = \frac{1}{N} \sum_{i=1}^{N} (\mathbf{M}_i - \mathbf{M}_i \cdot \mathbf{A}_i)^2, \tag{4}$$

where $\mathbf{M}$ denotes the binary foreground mask predicted by a state-of-the-art image matting model, and $\overline{\mathbf{A}}$ denotes the binarized version of the attention mask $\mathbf{A}$ computed between the suffix prompt tokens and the visual tokens extracted from the self-attention blocks within the diffusion transformer. $N$ denotes the number of pixels. In addition, we also use a trajectory magnitude to analyze whether the diffusion model is able to control the background region pixels across all timesteps throughout the entire denoising trajectory.

Figure 9 visualizes the attention maps between the suffix tokens and the visual tokens. We can see that by choosing a suitable suffix prompt, we can elicit the potential of the diffusion transformer to generate isolated background regions that are easy to segment.

**L. Effect of suffix prompt templates.** As shown in Table 9, the design of the suffix prompt is important for guiding the text-to-image generation models to generate images consisting of objects that can be easily isolated from the background by ensuring an approximately single-colored background.

| Suffix Prompt | Attention between Suffix text token and visual token | | | | Trajectory Magnitude | |
|---|---|---|---|---|---|---|
| | mIoU$_{BG}$ ↑ | mIoU$_{FG}$ ↑ | MSE$_{BG}$ ↓ | MSE$_{FGLeak}$ ↑ | $\bar{d}_{FG} - \bar{d}_{BG}$ ↑ | $\bar{d}_{BG}$ ↓ |
| original (w/o background prompt) | - | - | - | - | 0.041 | 6.198 |
| half green and half red background | 0.7863 | 0.5943 | 0.4717 | 0.2488 | -0.202 | 6.427 |
| half red and half blue background | 0.7318 | 0.5403 | 0.4868 | 0.2413 | -0.200 | 6.420 |
| half gray and half black background | 0.7902 | 0.5692 | 0.4478 | 0.2468 | 0.243 | 6.062 |
| half gray and half white background | 0.7787 | 0.5540 | 0.4701 | 0.2275 | 0.093 | 6.266 |
| a solid red background | 0.8282 | 0.6398 | 0.4414 | 0.2503 | -1.412 | 7.814 |
| a solid green background | 0.8554 | 0.6646 | 0.4706 | 0.2401 | -0.376 | 6.624 |
| a solid blue background | 0.8379 | 0.6493 | 0.4714 | 0.2416 | -0.485 | 6.818 |
| a solid black background | 0.7318 | 0.5179 | 0.4255 | 0.2409 | -1.749 | 8.317 |
| a solid white background | 0.8070 | 0.6495 | 0.3992 | 0.2365 | -2.503 | 9.083 |
| a solid transparent background | 0.5801 | 0.3302 | 0.4410 | 0.2262 | -1.413 | 7.872 |
| a solid gray background | 0.8642 | 0.6809 | 0.4181 | 0.2564 | 0.805 | 5.591 |

Table 9: Attention-map analysis of different suffix prompts.

| Method | Natural Object Layer Quality | | | Sticker Layer Quality | | | Creative Object Layer Quality | | |
|---|---|---|---|---|---|---|---|---|---|
| | HPSv2 ↑ | AE-V2.5 ↑ | TIPS ↑ | HPSv2 ↑ | AE-V2.5 ↑ | TIPS ↑ | HPSv2 ↑ | AE-V2.5 ↑ | TIPS ↑ |
| SuffixPrompt A | 26.13 | 5.609 | 29.83 | 26.07 | 4.758 | 25.67 | 29.12 | 5.572 | 36.25 |
| SuffixPrompt B | 26.29 | 5.587 | 29.95 | 25.98 | 4.726 | 25.45 | 29.28 | 5.529 | 36.32 |
| SuffixPrompt C | 26.32 | 5.625 | 30.06 | 26.14 | 4.758 | 25.77 | 29.35 | 5.566 | 36.42 |
| SuffixPrompt D | 25.95 | 5.631 | 29.65 | 26.23 | 4.745 | 25.93 | 29.38 | 5.539 | 36.12 |
| SuffixPrompt E | 26.07 | 5.493 | 29.35 | 26.12 | 4.739 | 25.76 | 28.78 | 5.497 | 34.84 |
| SuffixPrompt F | 26.01 | 5.607 | 29.43 | 26.10 | 4.755 | 25.75 | 29.28 | 5.518 | 35.70 |
| SuffixPrompt G | 26.45 | 5.468 | 30.07 | 25.72 | 4.654 | 25.30 | 29.87 | 5.397 | 36.14 |
| SuffixPrompt H | 26.58 | 5.617 | 30.19 | 26.14 | 4.735 | 25.69 | 29.55 | 5.551 | 36.25 |

Table 10: Effect of choosing different suffix prompt templates.

| Method | Natural Object Layer Quality | | | Sticker Layer Quality | | | Creative Object Layer Quality | | |
|---|---|---|---|---|---|---|---|---|---|
| | HPSv2 ↑ | AE-V2.5 ↑ | TIPS ↑ | HPSv2 ↑ | AE-V2.5 ↑ | TIPS ↑ | HPSv2 ↑ | AE-V2.5 ↑ | TIPS ↑ |
| Gray | 26.58 | 5.617 | 30.19 | 26.14 | 4.735 | 25.69 | 29.55 | 5.551 | 36.25 |
| Green | 25.59 | 5.304 | 28.72 | 25.62 | 4.605 | 25.02 | 28.78 | 5.342 | 34.52 |
| Blue | 26.29 | 5.434 | 29.53 | 25.83 | 4.690 | 25.63 | 29.29 | 5.456 | 35.55 |
| Red | 25.70 | 5.267 | 28.40 | 25.68 | 4.618 | 25.49 | 28.72 | 5.400 | 34.46 |
| White | 24.71 | 4.975 | 27.34 | 25.28 | 4.399 | 24.26 | 27.97 | 5.362 | 34.73 |
| Black | 26.16 | 5.500 | 29.38 | 25.34 | 4.655 | 24.96 | 28.78 | 5.430 | 34.48 |
| Transparent | 26.26 | 5.274 | 29.36 | 25.47 | 4.569 | 24.94 | 29.64 | 5.453 | 36.50 |
| Half green and half red | 25.91 | 5.344 | 29.03 | 25.93 | 4.699 | 26.08 | 29.72 | 5.399 | 35.79 |
| Half red and half blue | 25.83 | 5.418 | 29.10 | 25.99 | 4.691 | 26.05 | 29.75 | 5.459 | 35.89 |

Table 11: Effect of choosing different color within suffix prompt.

Here, we further compare the matting results of nine different suffix prompt designs in Table 10. We empirically find that choosing "*isolated on a solid gray background*" (SuffixPrompt H) achieves slightly better results.

**M. Effect of *color* within suffix prompt.** One natural question is which color is better for transparent layer generation. We investigate the influence of using different color words within the suffix prompt and summarize the results in Table 11. Accordingly, we find that using the color "gray" achieves the best results. This differs from the observation in previous work Burgert et al. (2024), which stated that using the color "green" performs best because "green" is the least common hue.

**N. Photorealistic multi-layer image synthesis.** In Figure 16, we adopt a top-down approach starting from the whole image generated by FLUX.1[dev] with synthetic object-driven prompts. Then, we train an anonymous object detector on our collected 800K multi-layer internal dataset to detect individual objects within the whole image. Following matting and inpainting, we obtain individual transparent layers and the background layer. After manual selection, we obtain 1K high-quality photo-realistic multi-layer images with accurate alpha mattes.

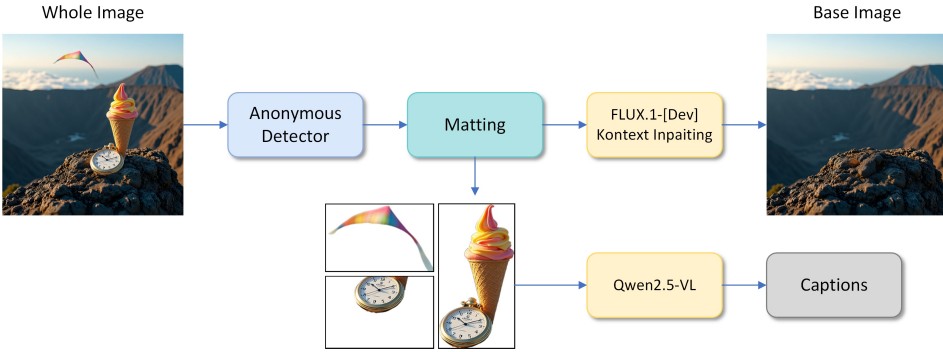

Figure 16: Photorealistic multi-layer data engine. Starting from a whole image generated by FLUX.1-[dev], we detect individual objects, and then leverage matting and inpainting to obtain individual transparent layers and the background layer.

**O. Qwen-Image Based Multi-layer Engine.** We further extent our method to Qwen-Image Wu et al. (2025), another state-of-the-art text-to-image generation model. As shown in Figure 21- 26, our method can also generate high-quality multi-layer transparent images based on Qwen-Image. We test first 50 images of the Japanese anime style split of PRISMLAYERSPRO, and the average TIPS score of all 491 layers of FLUX.1-[dev] and Qwen-Image are 22.53 and 23.45, respectively. Our TIPS model can effectively evaluate the quality of both text and object transparent layers.

**P. Advanced Photorealistic Multi-layer Data Engine.** As shown in Figure 17, we introduce our advanced photorealistic multi-layer data engine to further boost the performance of photorealistic multi-layer image generation. We mainly focus in improving the occlusion problem and physical consistency among different layers. We leverage Qwen-Image-Edit Wu et al. (2025) to regenerate the salient object in a blank background while keeping the original shape and details. By regenerating, we can overcome the occlusion problem and obtain high resolution layers with rich details while considering physical consistency.

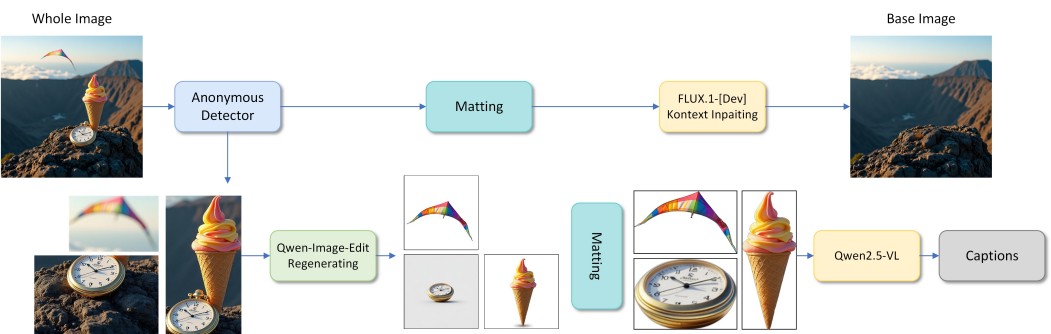

Figure 17: Advanced Photorealistic multi-layer data engine.

**Q. Analysis on Objects with Intrinsic Transparency** We conducted a dedicated evaluation focusing on the generation capabilities of our pipeline for objects with intrinsic transparency. We curated a list of 25 distinct object categories characterized by intrinsic transparency (e.g., glass marble, light bulb, ice cube). For each category, we leveraged Qwen3-VL-8B-Instruct Bai et al. (2025) to generate four distinct descriptive prompts, resulting in a total of 100 test prompts. We then utilized LayerFlux to generate the corresponding images in $1024 \times 1024$ resolution. For the segmentation, we evaluated two methods: RMBG-2.0 RMB and SAM 3 Carion et al. (2025). For SAM 3, the object category name was provided as the text prompt to guide the segmentation. Quantitative Evaluation: We assessed the quality of the segmented transparent regions using the TIPS (Transparency Information Performance Score) metric. We directly use the T2I prompt as the text description input to the TIPS model for evaluation. The evaluation was performed on the 100 generated samples.

- RMBG-2.0 achieved an average TIPS score of 26.78.
- SAM 3 achieved an average TIPS score of 24.15.

We provide all results and the corresponding prompts in the supplementary material. As shown in Figure 18, we visualized the segmentation results. We observed that handling complex transparency remains a challenging task for current state-of-the-art segmentation models. Specifically, both RMBG-v2.0 and SAM 3 struggle to perfectly disentangle semi-transparent regions or accurately recover fine-grained alpha values (opacity) in intricate areas. This limitation in high-fidelity alpha matting for transparent objects is acknowledged as an open problem and remains a subject for future work. However, despite these fine-grained limitations, our analysis demonstrates that for the majority of cases, the models successfully identify and segment the complete object mask (isolating the object from the background). This level of segmentation accuracy ensures that the generated assets are sufficiently high-quality and usable for the construction of our dataset, maintaining the integrity of the semantic layouts.

**R. Overlapping Layers** In Figure 19, we present the statistics of overlapping layers in our PRISMLAYERSPRO dataset. The average overlapping layer pair number is 6.32, indicating that our dataset contains diverse and complex multi-layer structures. As shown in Figure 20, we provide some examples

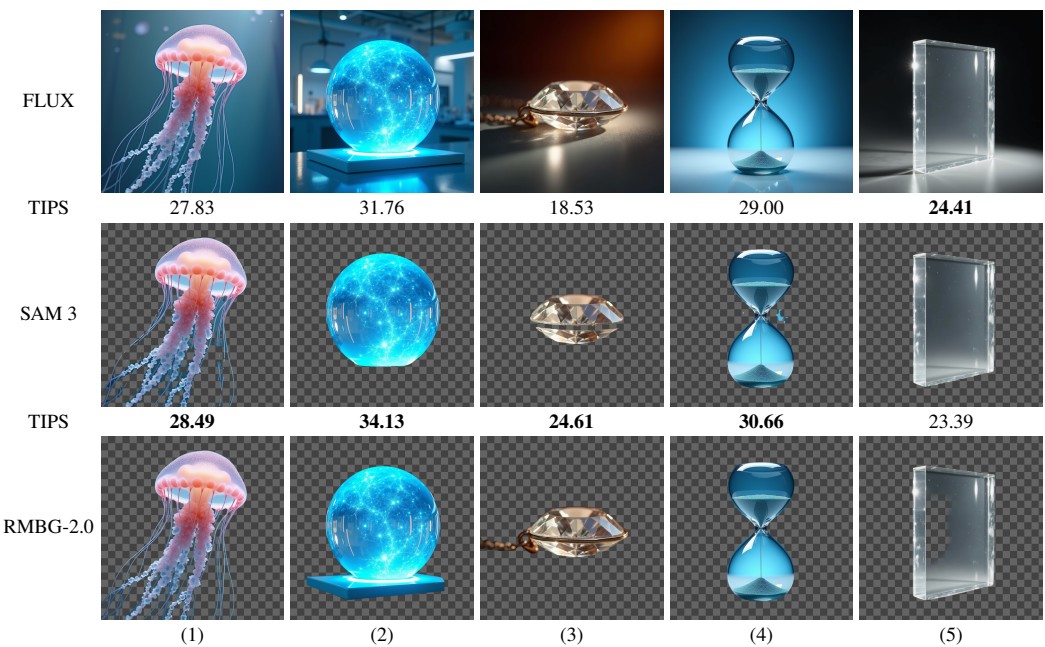

Figure 18: Qualitative comparison on objects with intrinsic transparency. From top to bottom, we show the generated image by FLUX.1-[dev], the segmentation results of SAM 3 and RMBG-2.0. The TIPS scores are shown below each result.

of overlapping layers in our PRISMLAYERSPRO dataset. We notice that the z-order of the layers is derived from real data and filtered by our proposed artifact detector and manual checking, ensuring the physical plausibility of our multi-layer images.

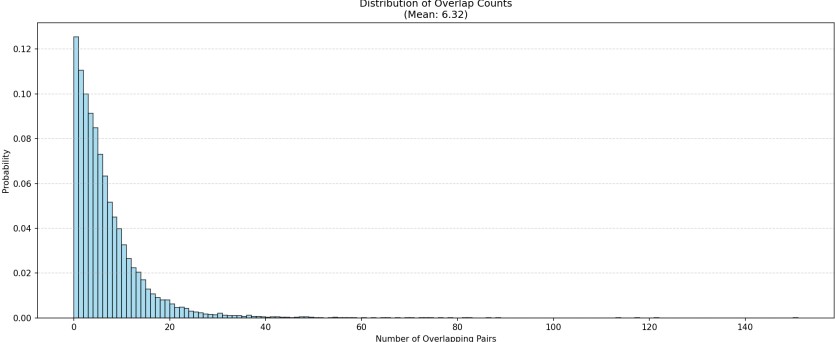

Figure 19: The statistics of overlapping layers in PRISMLAYERSPRO.

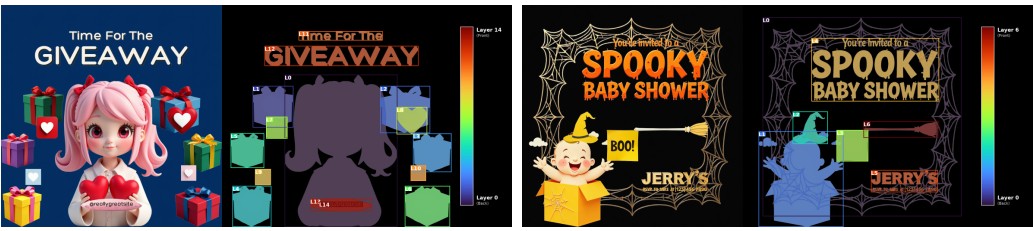

Figure 20: Examples of overlapping layers in our PRISMLAYERSPRO dataset.

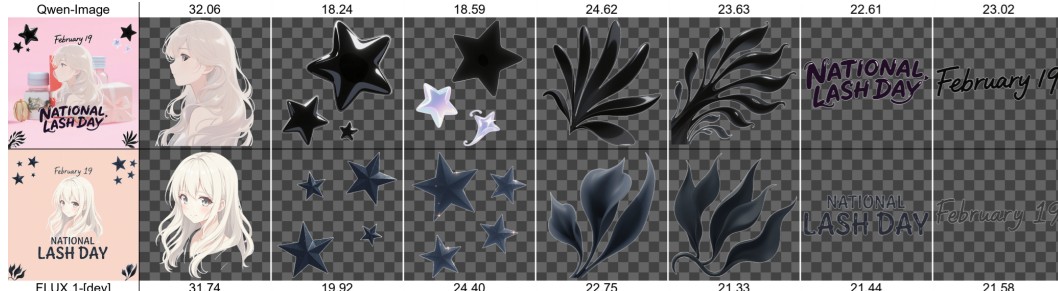

(1) This is a Japanese anime style image. The illustration features flowing, wavy hair with soft, delicate lines. The hair is depicted in a light shade, with subtle highlights that suggest a glossy texture. The strands cascade elegantly, framing the face and creating a sense of movement. The overall composition conveys a gentle, serene vibe, typical of anime aesthetics, emphasizing beauty and grace in the hair's design, style: elegant, harmonious, sophisticated, isolated on a solid gray background.

(2) This is a Japanese anime style image. Text: None Three sparkling star shapes float gracefully, each varying in size and orientation. The largest star, with smooth, curvy edges, radiates a sense of whimsy, while the medium star has a slightly sharper silhouette, adding dynamic contrast. The smallest star, delicate and petite, completes the trio. Each star is a deep, rich black, exuding a glossy sheen that suggests a magical aura, enhancing the enchanting atmosphere of the composition, style: elegant, harmonious, sophisticated, isolated on a solid gray background.

(3) This is a Japanese anime style image. Three sparkling star shapes float gracefully, each varying in size and orientation. The largest star, a bold black silhouette, radiates a sense of mystery, while the medium star has a softer curve, suggesting a gentle twinkle. The smallest star, with a playful tilt, adds a whimsical touch. Their smooth, glossy surfaces reflect light, enhancing the enchanting atmosphere, reminiscent of magical moments in anime, style: elegant, harmonious, sophisticated, isolated on a solid gray background.

(4) This is a Japanese anime style image. The image features a dynamic arrangement of five stylized, elongated black shapes resembling fingers or petals, radiating outward from a corner. Each shape has a smooth, glossy texture, with subtle highlights that suggest a soft sheen. The forms vary in size and curvature, creating a sense of movement and flow, as if they are gently unfurling or reaching out, evoking a feeling of grace and elegance, style: elegant, harmonious, sophisticated, isolated on a solid gray background.

(5) This is a Japanese anime style image. The composition features a series of stylized, black, organic shapes resembling flowing, elongated fingers or petals. Each shape has a smooth, glossy texture, with subtle gradients that suggest depth and curvature. The arrangement creates a dynamic sense of movement, as if the forms are gently reaching outward, inviting the viewer into a whimsical, fantastical realm. The interplay of light and shadow enhances the overall elegance of the design, style: elegant, harmonious, sophisticated, isolated on a solid gray background.

(6) Text: 'NATIONAL LASH DAY' in a playful, curved font, featuring bold, dark letters with a slight glossy sheen, reminiscent of a whimsical anime title. The letters are slightly uneven, adding a charming, hand-drawn quality, evoking a sense of celebration and joy, style: elegant, harmonious, sophisticated, isolated on a solid gray background.

(7) Text: 'February 19' in playful, handwritten black script with a slight tilt, evoking a sense of whimsy and charm, reminiscent of a light-hearted Japanese anime scene, style: elegant, harmonious, sophisticated, isolated on a solid gray background.

Figure 21: Qualitative comparison of multi-layer transparent image generation based on Qwen-Image and FLUX.1-[dev]. The TIPS scores are shown above and below each layer.

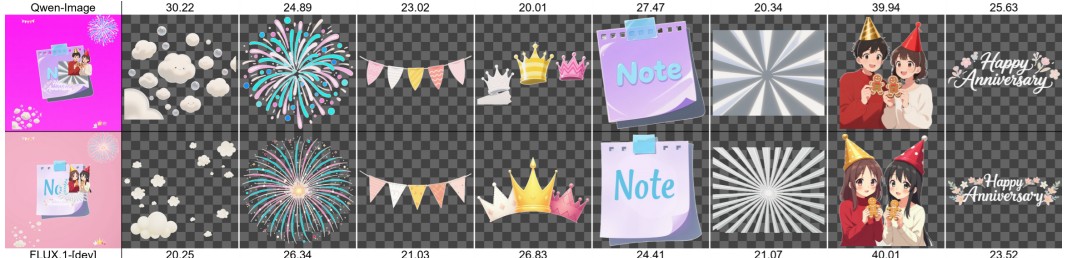

(1) This is a Japanese anime style image. The foreground features a whimsical scattering of soft, rounded shapes in varying sizes, resembling delicate, fluffy clouds or playful bubbles. Each shape is a gentle off-white, with subtle shading that gives them a three-dimensional appearance. The arrangement is random yet harmonious, creating a sense of lightness and joy, as if they are floating in a serene, dreamlike atmosphere, style: elegant, harmonious, sophisticated, isolated on a solid gray background.

(2) This is a Japanese anime style image. Text: None A vibrant explosion of color radiates outward, featuring delicate, swirling lines in bright turquoise and soft pink. The lines, resembling fireworks, are adorned with small, circular accents in varying shades of blue and pink, creating a whimsical, dynamic effect. The intricate layering of the lines suggests movement, as if the fireworks are bursting into life, illuminating the scene with a playful, enchanting energy, style: elegant, harmonious, sophisticated, isolated on a solid gray background.

(3) This is a Japanese anime style image. A whimsical string of five colorful triangular flags flutters gently, each adorned with unique patterns. The first flag is a soft pink with delicate zigzag lines, followed by a white flag featuring subtle stripes. Next, a vibrant yellow flag stands out with bold diagonal lines, while a coral flag showcases a playful wavy design. The final flag is a light pastel pink, completing the cheerful display, style: elegant, harmonious, sophisticated, isolated on a solid gray background.

(4) This is a Japanese anime style image. Three whimsical crowns rise playfully along a curved horizon. The leftmost crown is a soft white with delicate, wavy patterns, while the center crown shines in a bright yellow, adorned with bold, horizontal stripes. The rightmost crown is a vibrant pink, featuring intricate zigzag designs. Each crown has a slight sheen, suggesting a magical aura, and they are layered in a way that creates a sense of depth and movement, style: elegant, harmonious, sophisticated, isolated on a solid gray background.

(5) This is a Japanese anime style image. Text: 'Note' in playful, rounded, pastel blue lettering with a soft shadow effect. A large, lavender square note paper dominates the scene, featuring a glossy texture that catches light. At the top, a translucent, sky-blue tape holds the note, adding a whimsical touch. The note's edges are slightly curled, suggesting a gentle breeze, while small, square cutouts along the top create a charming, decorative border, style: elegant, harmonious, sophisticated, isolated on a solid gray background.

(6) This is a Japanese anime style image. Text: None. The image features a dynamic arrangement of elongated, angular lines radiating outward, resembling sun rays. Each line is a soft white, with subtle gradients that suggest depth. The varying thicknesses create a sense of movement, as if the rays are bursting forth. The interplay of light and shadow adds a delicate texture, enhancing the overall energetic feel of the composition, style: elegant, harmonious, sophisticated, isolated on a solid gray background.

(7) This is a Japanese anime style image. The scene features two individuals wearing festive party hats shaped like cones, one in gold and the other in a vibrant red. They hold gingerbread cookies shaped like cheerful figures, with intricate icing details. The textures of their clothing, a cozy red sweater and a soft white top, contrast beautifully. The warm colors and playful expressions evoke a joyful celebration, enhanced by the whimsical atmosphere typical of anime, style: elegant, harmonious, sophisticated, isolated on a solid gray background.

(8) Text: 'Happy Anniversary' in elegant, flowing white script with a soft, shimmering effect, surrounded by delicate pastel-colored floral motifs that evoke a sense of celebration and warmth. The letters appear to dance lightly, as if caught in a gentle breeze, enhancing the joyful atmosphere of the occasion, style: elegant, harmonious, sophisticated, isolated on a solid gray background.

Figure 22: Qualitative comparison of multi-layer transparent image generation based on Qwen-Image and FLUX.1-[dev]. The TIPS scores are shown above and below each layer.

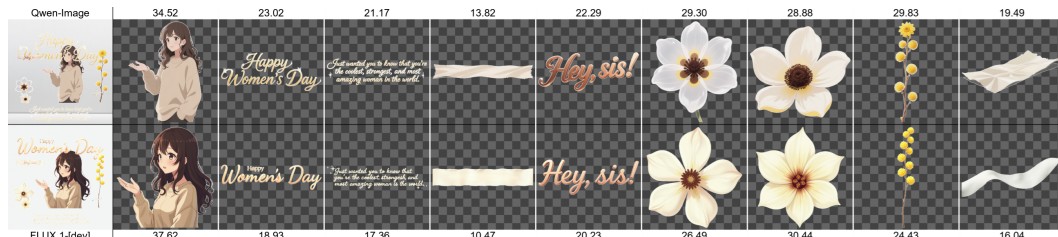

(1) This is a Japanese anime style image. A young woman with flowing, wavy hair, dressed in a cozy, oversized beige sweater, gestures with her right hand, palm up, as if inviting conversation. The soft texture of her sweater contrasts with the smooth, light wood paneling in the background. A stylish white lamp stands nearby, casting a warm glow, enhancing the inviting atmosphere of the scene, style: elegant, harmonious, sophisticated, isolated on a solid gray background.

(2) Text: 'Happy Women's Day' in elegant, flowing cursive with a shimmering golden hue, each letter adorned with delicate highlights that evoke a sense of warmth and celebration, reminiscent of a gentle breeze in a serene anime landscape. The letters dance gracefully, embodying the spirit of joy and empowerment, inviting all to partake in the festivities of this special day, style: elegant, harmonious, sophisticated, isolated on a solid gray background.

(3) Text: 'Just wanted you to know that you're the coolest, strongest, and most amazing woman in the world.' in elegant, flowing cursive with a soft golden hue, each letter shimmering as if kissed by starlight, surrounded by delicate sparkles that evoke a sense of warmth and admiration, reminiscent of heartfelt messages in a Japanese anime scene, style: elegant, harmonious, sophisticated, isolated on a solid gray background.

(4) This is a Japanese anime style image. The image features a long, narrow strip of soft, creamy beige fabric with a slightly frayed edge, suggesting a delicate texture. The fabric appears to have a subtle sheen, reflecting light gently, enhancing its ethereal quality. The strip is positioned horizontally, evoking a sense of calm and simplicity, reminiscent of traditional Japanese textiles, with a serene and understated elegance, style: elegant, harmonious, sophisticated, isolated on a solid gray background.

(5) Text: 'Hey, sis!' in elegant, flowing cursive with a warm, bronze hue, featuring a soft, shimmering effect that gives it a three-dimensional appearance, as if the letters are gently rising from the surface. The text is slightly tilted, adding a playful dynamic, and the subtle highlights enhance its charm, reminiscent of a whimsical Japanese anime aesthetic, style: elegant, harmonious, sophisticated, isolated on a solid gray background.

(6) This is a Japanese anime style image. The delicate flower features five translucent white petals, each with soft, rounded edges and a subtle sheen that catches the light. The petals are adorned with faint yellow and brown accents near the base, creating a gentle gradient. At the center, a rich, dark brown cluster of stamen radiates outward, adding depth and contrast. The overall composition exudes a serene, ethereal beauty, reminiscent of a tranquil anime garden scene, style: elegant, harmonious, sophisticated, isolated on a solid gray background.

(7) This is a Japanese anime style image. A delicate flower with five soft, creamy white petals radiates outward, each petal slightly curled at the edges, showcasing a subtle sheen. The center is a rich, dark brown, densely packed with tiny, intricate stamens, creating a striking contrast. The petals exhibit gentle gradients, hinting at a whisper of yellow near the base, enhancing their ethereal beauty. The overall composition evokes a sense of tranquility and grace, style: elegant, harmonious, sophisticated, isolated on a solid gray background.

(8) This is a Japanese anime style image. A slender, delicate branch stretches vertically, adorned with vibrant, fluffy yellow pom-pom flowers that resemble miniature suns. Each flower is a bright, cheerful yellow, with soft, feathery textures that catch the light, creating a gentle glow. The thin, slightly twisted stem is a muted brown, adding a natural contrast to the vivid blossoms, evoking a sense of serene beauty and whimsy, style: elegant, harmonious, sophisticated, isolated on a solid gray background.

(9) This is a Japanese anime style image. A delicate, crumpled strip of pale cream paper lies at an angle, its soft texture resembling silk. Subtle shadows accentuate its folds, creating a sense of depth. The edges are slightly frayed, hinting at gentle wear. The light catches the surface, giving it a faint sheen, while the overall composition evokes a serene, minimalist aesthetic typical of anime art, style: elegant, harmonious, sophisticated, isolated on a solid gray background.

Figure 23: Qualitative comparison of multi-layer transparent image generation based on Qwen-Image and FLUX.1-[dev]. The TIPS scores are shown above and below each layer.

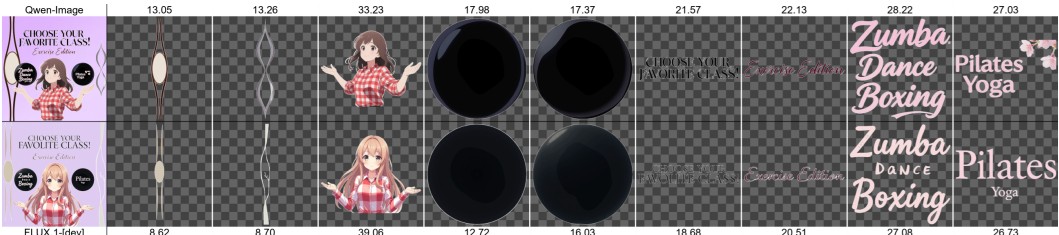

(1) This is a Japanese anime style image. The design features two elongated, parallel lines that gracefully extend vertically, converging at a central, symmetrical oval shape. The lines are outlined in a deep, rich brown, evoking a sense of elegance. The oval, nestled between the lines, adds a touch of softness, creating a harmonious balance. The overall composition suggests a serene, minimalist aesthetic, reminiscent of traditional Japanese art forms, style: elegant, harmonious, sophisticated, isolated on a solid gray background.

(2) This is a Japanese anime style image. The design features two elongated, parallel lines that gracefully converge at the center, forming a delicate, symmetrical shape reminiscent of an hourglass. The lines are outlined in a soft, dark hue, creating a subtle contrast against the background. The central shape, an elegant oval, adds a sense of harmony and balance, evoking a feeling of tranquility and fluidity in its minimalist elegance, style: elegant, harmonious, sophisticated, isolated on a solid gray background.

(3) This is a Japanese anime style image. The character wears a red and white checkered shirt, with soft, flowing hair cascading down her shoulders. Her arms are outstretched, palms up, suggesting a welcoming gesture. The shirt's fabric appears textured, with subtle shading enhancing its folds. The overall composition conveys a sense of openness and curiosity, embodying the vibrant and expressive nature typical of anime art, style: elegant, harmonious, sophisticated, isolated on a solid gray background.

(4) This is a Japanese anime style image. A perfectly round, deep black circle dominates the scene, exuding an aura of mystery and elegance. The surface appears smooth and glossy, reflecting subtle hints of light that create a soft sheen. The edges are sharply defined, enhancing the contrast against the surrounding space, while the overall simplicity invites intrigue, reminiscent of a portal to another world in a fantastical anime setting, style: elegant, harmonious, sophisticated, isolated on a solid gray background.

(5) This is a Japanese anime style image. A large, perfectly round shape fills the frame, rendered in a deep, rich black hue. The surface appears smooth and glossy, reflecting subtle highlights that suggest a soft sheen. The edges are sharply defined, creating a striking contrast against the surrounding space, evoking a sense of depth and dimensionality, as if the circle is floating in an ethereal realm, style: elegant, harmonious, sophisticated, isolated on a solid gray background.

(6) Text: 'CHOOSE YOUR FAVORITE CLASS!' in bold, elegant black lettering with a slight glossy sheen, exuding a sense of adventure and excitement, reminiscent of a vibrant anime world where choices lead to epic journeys, style: elegant, harmonious, sophisticated, isolated on a solid gray background.

(7) Text: 'Exercise Edition' in elegant, slender, dark maroon lettering, exuding a soft, refined aura, reminiscent of a serene Japanese calligraphy style, style: elegant, harmonious, sophisticated, isolated on a solid gray background.

(8) Text: 'Zumba' in soft pink, elegant font, 'Dance' in the same delicate style, and 'Boxing' in matching pastel pink, all aligned vertically with a gentle, flowing rhythm, evoking a sense of movement and energy, reminiscent of a lively anime dance scene, style: elegant, harmonious, sophisticated, isolated on a solid gray background.

(9) Text: 'Pilates' in soft pink, elegant, rounded font, and 'Yoga' in matching soft pink, slightly larger, both exuding a calming, serene aura reminiscent of gentle cherry blossoms in spring, style: elegant, harmonious, sophisticated, isolated on a solid gray background.

Figure 24: Qualitative comparison of multi-layer transparent image generation based on Qwen-Image and FLUX.1-[dev]. The TIPS scores are shown above and below each layer.

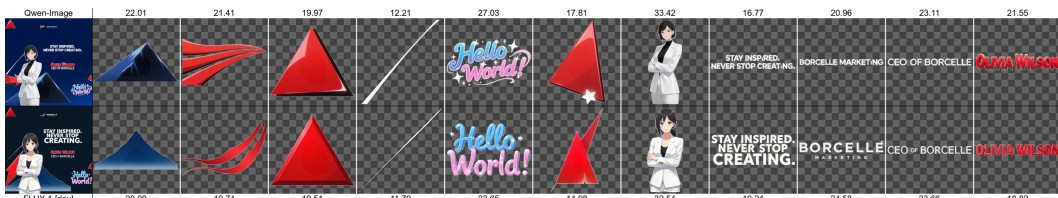

(1) This is a Japanese anime style image. The image features a large, dark blue triangular shape, resembling a stylized roof or mountain peak. The edges are sharply defined, creating a bold silhouette. The surface has a smooth texture, reflecting light subtly, enhancing its dimensionality. The overall composition conveys a sense of stability and strength, with the triangular form suggesting upward movement, evoking feelings of aspiration and adventure, style: elegant, harmonious, sophisticated, isolated on a solid gray background.

(2) This is a Japanese anime style image. Three sleek, elongated red stripes curve gracefully, each one slightly overlapping the next. The vibrant red hue is accentuated by a subtle black outline, giving the shapes a bold, dynamic appearance. The stripes taper at one end, suggesting a sense of motion, as if they are flowing or being swept by a gentle breeze, embodying a sense of energy and fluidity, style: elegant, harmonious, sophisticated, isolated on a solid gray background.

(3) This is a Japanese anime style image. A vibrant red triangle dominates the composition, its sharp edges contrasting against a subtle black outline that adds depth. The surface of the triangle exhibits a smooth texture, reflecting light in a way that suggests a glossy finish. The bold color evokes a sense of energy and passion, while the geometric simplicity creates a striking visual impact, reminiscent of dynamic anime aesthetics, style: elegant, harmonious, sophisticated, isolated on a solid gray background.

(4) This is a Japanese anime style image. The image features a single, slender white line that diagonally traverses the canvas, creating a sense of dynamic movement. The line is smooth and sharp, with a slight glow that suggests an ethereal quality. Its placement draws the eye across the space, evoking a feeling of tension and elegance, reminiscent of minimalist anime aesthetics, style: elegant, harmonious, sophisticated, isolated on a solid gray background.

(5) This is a Japanese anime style image. Text: 'Hello World!' in vibrant, swirling blue and pink letters, adorned with sparkling stars and a soft, ethereal glow, creating a whimsical and inviting atmosphere. The letters are playfully arranged, with the 'H' and 'W' slightly larger, giving a dynamic feel, as if they are dancing across the canvas, inviting viewers into a magical realm, style: elegant, harmonious, sophisticated, isolated on a solid gray background.

(6) This is a Japanese anime style image. A vibrant red triangle, sharply pointed to the left, dominates the scene. Its edges are outlined in a thin, dark line, enhancing its boldness. The surface of the triangle has a smooth texture, reflecting light subtly, giving it a slightly glossy appearance. A small white star glimmers at the base, adding a touch of whimsy and charm to the overall composition, style: elegant, harmonious, sophisticated, isolated on a solid gray background.

(7) This is a Japanese anime style image. The figure is dressed in a sleek, tailored white blazer, exuding an air of confidence. The fabric appears smooth with subtle highlights, reflecting light elegantly. The blazer is complemented by a black top underneath, creating a striking contrast. The arms are crossed, suggesting a poised demeanor. The overall composition conveys a sense of professionalism and strength, enhanced by the clean lines and polished appearance of the attire, style: elegant, harmonious, sophisticated, isolated on a solid gray background.

(8) Text: "STAY INSPIRED. NEVER STOP CREATING." in large, bold, white lettering with a soft, ethereal glow, reminiscent of a magical aura, embodying the spirit of creativity and motivation in a vibrant, anime-inspired style, style: elegant, harmonious, sophisticated, isolated on a solid gray background.

(9) Text: 'BORCELLE MARKETING' in sleek, modern sans-serif font, with a crisp white color that stands out prominently, evoking a sense of professionalism and clarity. The letters are evenly spaced, creating a balanced composition that draws the eye effortlessly across the text. The overall aesthetic is clean and minimalistic, embodying a contemporary design that resonates with the themes of innovation and efficiency, reminiscent of a Japanese anime style, style: elegant, harmonious, sophisticated, isolated on a solid gray background.

(10) Text: 'CEO OF BORCELLE' in sleek, modern, sans-serif white lettering, exuding a professional aura, with a subtle shadow effect that adds depth, reminiscent of a high-tech corporate environment in a Japanese anime style, style: elegant, harmonious, sophisticated, isolated on a solid gray background.

(11) Text: 'OLIVIA WILSON' in bold, vibrant red lettering with a slight 3D effect, giving it a dynamic presence. The letters are sharply defined, with a glossy finish that reflects light, enhancing their prominence. The overall composition exudes a sense of energy and style, reminiscent of a character nameplate in a Japanese anime series, capturing attention with its striking color and bold typography, style: elegant, harmonious, sophisticated, isolated on a solid gray background.

Figure 25: Qualitative comparison of multi-layer transparent image generation based on Qwen-Image and FLUX.1-[dev]. The TIPS scores are shown above and below each layer.

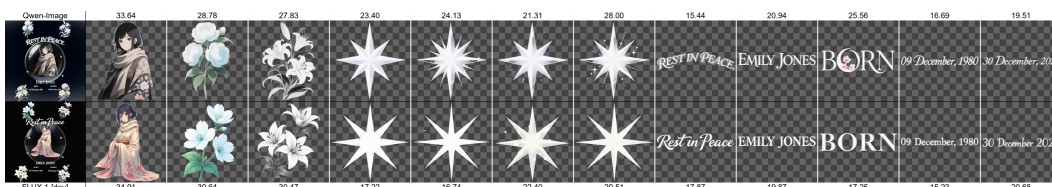

(1) This is a Japanese anime style image. The figure is adorned in a richly patterned shawl featuring delicate floral motifs in soft pastels, contrasting with the deep shadows around. The intricate textures of the fabric suggest warmth and comfort. A flowing scarf wraps around the neck, cascading down with gentle folds, while the overall composition evokes a sense of mystery and depth, enhanced by the elegant arch framing the figure, style: elegant, harmonious, sophisticated, isolated on a solid gray background.

(2) This is a Japanese anime style image. Text: None. Delicate, large blossoms with rounded petals in soft shades of white and pale blue, intricately detailed with fine lines. The flowers are adorned with subtle highlights, giving them a gentle glow. Graceful green leaves with intricate veins cascade alongside, enhancing the floral arrangement's elegance. A slender stem supports the blooms, creating a harmonious flow, evoking a sense of serene beauty and tranquility typical of anime aesthetics, style: elegant, harmonious, sophisticated, isolated on a solid gray background.

(3) This is a Japanese anime style image. Text: None. The artwork features a delicate arrangement of elegant lilies, their petals intricately detailed with soft curves and sharp edges. The flowers are depicted in a monochromatic scheme, showcasing a blend of smooth and textured surfaces. Wispy leaves intertwine gracefully, enhancing the composition's fluidity. The overall design exudes a serene beauty, reminiscent of traditional Japanese art, with a focus on harmony and natural elegance, style: elegant, harmonious, sophisticated, isolated on a solid gray background.

(4) This is a Japanese anime style image. A radiant, eight-pointed star glimmers at the center, its sharp, elongated points radiating outward. The star is a brilliant white, with a soft, ethereal glow that creates a shimmering effect around its edges. Subtle gradients of light and shadow enhance its three-dimensional appearance, giving it a magical, otherworldly quality, reminiscent of celestial phenomena in anime, style: elegant, harmonious, sophisticated, isolated on a solid gray background.

(5) This is a Japanese anime style image. A radiant starburst glimmers at the center, featuring sharp, elongated points that radiate outward. The star is a brilliant white, with a soft, ethereal glow that creates a shimmering effect. Each point is delicately outlined in a subtle silver hue, enhancing its luminous quality. The interplay of light and shadow gives the star depth, evoking a sense of magic and wonder, style: elegant, harmonious, sophisticated, isolated on a solid gray background.

(6) This is a Japanese anime style image. A radiant, eight-pointed star glimmers with a soft, ethereal glow at its center, surrounded by delicate, shimmering rays that extend outward. The star's points are sharp and elongated, each adorned with a subtle gradient from bright white at the core to a gentle silver at the tips, creating a mesmerizing effect. The overall composition evokes a sense of magic and wonder, reminiscent of celestial themes in anime, style: elegant, harmonious, sophisticated, isolated on a solid gray background.

(7) This is a Japanese anime style image. A radiant starburst glimmers at the center, featuring sharp, elongated points that radiate outward. The star is a brilliant white, with soft gradients transitioning to a delicate silver at the tips. Subtle sparkles surround it, enhancing its ethereal glow. The overall effect is one of magical luminescence, evoking a sense of wonder and enchantment, typical of anime aesthetics, style: elegant, harmonious, sophisticated, isolated on a solid gray background.

(8) Text: 'REST IN PEACE' in elegant, flowing white lettering, arched gracefully, reminiscent of a serene memorial. The font features soft curves and sharp edges, embodying a tranquil yet poignant atmosphere, enhanced by a subtle shadow that adds depth and dimension. The overall composition evokes a sense of calm and reflection, perfectly suited for a Japanese anime style, where emotions are conveyed through delicate artistry, style: elegant, harmonious, sophisticated, isolated on a solid gray background.

(9) Text: 'EMILY JONES' in large, bold, elegant white lettering with a soft, ethereal glow, reminiscent of a magical aura, embodying a serene and enchanting atmosphere typical of Japanese anime, style: elegant, harmonious, sophisticated, isolated on a solid gray background.

(10) Text: 'BORN' in elegant, slender, white lettering with a soft, ethereal glow, reminiscent of moonlight filtering through cherry blossoms, embodying a sense of new beginnings and hope, style: elegant, harmonious, sophisticated, isolated on a solid gray background.

(11) Text: '09 December, 1980' in elegant, soft white lettering with a delicate, shimmering effect, reminiscent of moonlight, evoking a sense of nostalgia and tranquility. The characters are gracefully spaced, with a slight curve that adds a whimsical touch, embodying the charm of Japanese anime aesthetics, style: elegant, harmonious, sophisticated, isolated on a solid gray background.

(12) Text: '30 December, 2022' in elegant, soft white lettering with a delicate, shimmering effect, reminiscent of moonlight reflecting on water, embodying a serene and tranquil atmosphere typical of Japanese anime aesthetics, style: elegant, harmonious, sophisticated, isolated on a solid gray background.

Figure 26: Qualitative comparison of multi-layer transparent image generation based on Qwen-Image and FLUX.1-[dev]. The TIPS scores are shown above and below each layer.

### Text Sticker Recaption Prompt for GPT-4o

You are given the key word of a text sticker and its corresponding image. Your task is to generate an accurate and descriptive caption for the sticker, following these guidelines:
1. The caption begins with "The text sticker describes/contains/" and ends with "isolated on a solid transparent background."
2. Clearly describe the text in the sticker, including the font color, font style, and any visual effects (e.g., shadows, gradients) observed in the image.
3. Keywords usually refer to the text in the sticker, and you may include other relevant descriptive elements. Be explicit about these in your caption.
4. Refer to the examples provided for clarity on how to construct your caption. Aim for creativity while adhering to the required structure.
Here are some examples for reference:
- "The text sticker presents the word 'Focus' in a sharp, modern font, filled with a gradient of charcoal gray to bright red. The letters are outlined in bright white, and stylized targets surround the text, conveying determination and clarity, isolated on a solid transparent background."
- "The text sticker showcases the word 'Celebrate' in a festive, curly font, filled with a vibrant confetti gradient of rainbow colors. Each letter is dotted with tiny sparkles, and balloons and streamers float around, enhancing the joyful spirit of celebration, isolated on a solid transparent background."
Please ensure to generate a caption that fits this style and adheres to the guidelines.
\*\*\*\*\*\*\*\*\*\*\*\*\*\*\*\*\*\*\*\*\*\*\*\*\*\*\*\*\*\*\*\*\*\*\*\*\*\*\*\*\*\*\*\*\*\*\*\*\*\*\*\*
**Response 1**:
{response 1}
\*\*\*\*\*\*\*\*\*\*\*\*\*\*\*\*\*\*\*\*\*\*\*\*\*\*\*\*\*\*\*\*\*\*\*\*\*\*\*\*\*\*\*\*\*\*\*\*\*\*\*\*
Please strictly follow the following format requirements when outputting, and don't have any other unnecessary words.
**Output Format**:
response 1 or response 2.

### Creative Object Layer Prompt for GPT-4o

You are tasked with generating imaginative and creative image descriptions based on a given object word. The generated description should follow these specific guidelines:
### **1. Input:**
- You will receive a single object word (e.g., "penguin", "teapot", "robot", etc.).
- Use this object as the central focus of the description.
### **2. Output Requirements:**
- The description should be **creative and unexpected**, modifying the object or adding elements that make it unusual, humorous, or visually striking.
- The description **must not include details about the background**—focus only on the main object and any additional elements that make it more interesting.
- Aim for a **concise but vivid** description, ideally **within 20 to 30 words**.
- Use **strong visual language** to create a mental image.
- Avoid generic descriptions—make it **fun, unique, and imaginative**.
### **3. Examples for Reference:**
| Given Object | Generated Description |
|————|—————|
| Kangaroo | A kangaroo holding a beer, wearing ski goggles and passionately singing silly songs. |
| Car | A car made out of vegetables. |
| Raccoon | A cyberpunk-styled raccoon wearing neon glasses and a futuristic jacket, holding a laser gun in one paw. |
| Teapot | A giant teapot with robotic arms, serving tea while wearing a tiny monocle and top hat. |
| Penguin | A punk-styled penguin with a mohawk, leather jacket, and electric guitar, rocking out on an ice stage. |
### **4. Constraints & Guidelines:**
- Do **not** include the background in the description.
- Feel free to **modify the object's appearance, abilities, or accessories** to make it more interesting.
- If necessary, **add related objects** (e.g., a robot might have futuristic gadgets, a dog might have sunglasses and a skateboard).
- Keep the tone fun, artistic, and engaging.
### **5. Additional Notes:**
Please directly respond to the prompt with the creative description.

**Multi-layer Style Recaption Instruction for GPT-4o**

You will receive an RGBA image placed on a gray background. Your task is to generate a highly detailed description of the image's content while adhering to a given stylistic (STYLEPROMPT) requirement.

**Key Guidelines:**

1. **Ignore the Gray Background:** - Do not mention or describe the gray background in any way. Focus solely on the foreground content.

2. **Handling Text in the Image:** - If the image contains any textual elements, the description **must** begin with **"Text:"** followed by a precise transcription of all visible text. - Transcribe every word, symbol, punctuation mark, and character **without omission or modification**. - The description of text must be brief and the style description should be limited to 5 words.

3. **Handling Non-Text Elements:** - If the image contains **non-text elements**, generate an **detailed** description, capturing all visible aspects. - Ensure that the provided style, STYLEPROMPT, is seamlessly **integrated into the description**, maintaining coherence and natural flow.

4. **Output Format:** - Provide only the description of the image. Do **not** include any additional explanations, comments, or meta-information about the task itself. - The description **must explicitly state** that the image is in **STYLEPROMPT style**, starting with **"This is a STYLEPROMPT style image."** (VERY IMPORTANT) - Limited to 70 words!!!

The image is shown below:

