# OpenReview forum: "PrismLayers: Open Data for High-Quality Multi-Layer Transparent Image Generative Models"
_ICLR.cc/2026/Conference — Submitted to ICLR 2026_

### Official Review · Reviewer_H3ue · 2025-10-16

**Soundness:** 2
**Presentation:** 2
**Contribution:** 3
**Rating:** 4
**Confidence:** 4

**Summary:**

The paper releases PrismLayers family datasets for multi-layer transparent image generation and builds LayerFLUX / MultiLayerFLUX, plus an ART+ model fine-tuned on the data. Most pipelines and benchmarks are coupled with FLUX.1-[dev].

**Strengths:**

- Fills a data gap for layered images with alpha mattes; gives sizes and curation steps.
- Clear pipeline for generating layers then composing; user studies and metrics are reported.

**Weaknesses:**

- Possible double-blind violation: page-1 public Hugging Face link to the dataset; this can reveal author identity.
- Styles skew to design/cartoon/3D (e.g., toy, ink, doodle). The photoreal set is only 1K; real-image diversity is limited.
- Fig. 3 text/typography is hard to read; legends could be clearer.
- Base-model narrowness: training-free pipeline and benchmarks are largely FLUX-centric; generalization to other backbones is not shown.

**Questions:**

Can you test other base models (e.g., SDXL/DiT/flow variants) to show model-agnostic value beyond FLUX?

---

> ### Author Response · Authors · 2025-11-23
>
> We appreciate your thoughtful review and your recognition of the data gap we address, as well as the clarity of the pipeline and experiments. Below, we summarize our understanding of your comments and respond to your concerns.
>
> **Strengths you highlighted**
> - Our work fills a data gap for layered images with alpha mattes and clearly describes sizes and curation steps.
> - The pipeline for generating layers and composing them (LayerFLUX + MultiLayerFLUX) is clearly presented; user studies and metrics are reported.
>
> **Main weaknesses / concerns**
> 1. A potential double-blind violation due to a public Hugging Face link on page 1.
> 2. Style skew towards design/cartoon/3D (toy, ink, doodle, etc.); the photorealistic set is limited.
> 3. Figure 3’s text/typography is hard to read; legends could be clearer.
> 4. The pipeline and benchmarks are heavily FLUX-centric, and generalization to other backbones is not shown.
>
> **On the double-blind issue**
>
> We sincerely apologize for including a public Hugging Face URL on page 1. The repository itself does not contain author names, affiliations, or other identifying metadata. However, we fully understand that an external URL in the main paper can still be perceived as a violation.
> In the revised version, we
> - Remove the explicit URL from the main paper
>
> **Style skew and limited real-image diversity**
>
> You are correct that PrismLayersPro and the main synthetic datasets skew toward design/cartoon/3D styles (toy, ink, doodle, melting silver, etc.), as also shown by the style distribution in Figure 1. This is intentional in the sense that our target is graphic design workflows, but we agree that it limits diversity in the purely photorealistic regime.
> As discussed in our responses to other reviewers, we:
> - clarify the scope of the introduction and discussion:
>   - the primary purpose of PrismLayers is to support editable design-like multi-layer images, not full physical photo realism.
> - better describe PrismLayersReal as a small but high-quality photorealistic subset constructed via the top-down pipeline in Appendix N
> - outline our ongoing plan to increase the size and diversity of PrismLayersReal, by:
>   - running the same pipeline on a larger pool of FLUX and real photographs,
>   - expanding the object detector’s category set, and
>   - systematically curating photo-centric scenes (e.g., indoor/outdoor, humans, everyday objects).
>
> **Figure 3: readability and legends**
>
> We agree that Figure 3 is dense, and typography can be hard to read.
> In the revised version, we rewrote the figure caption to clarify the legends.
>
> **FLUX-centric evaluation and other base models**
>
> We chose FLUX.1-[dev] as our main base model because it is a strong, publicly available diffusion transformer with excellent text alignment and aesthetics, making it an attractive testbed. We agree that showing results on other backbones would strengthen the claim that LayerFLUX / MultiLayerFLUX are model-agnostic.
> During the discussion period, we will:
> - Run LayerFLUX + MultiLayerFLUX with one additional T2I model (Qwen-Image) in Appendix O, using the same suffix-prompt and matting strategy.
> - Evaluate on a subset of Layer-Bench and our internal multi-layer prompts, reporting:
>   - TIPS and aesthetic scores for single-layer transparent outputs,
>   - qualitative examples showing that the suffix-prompt and matting scheme still produce clean alpha mattes.

---

> > ### Comment · Reviewer_H3ue · 2025-11-25
> >
> > My core concerns remain unresolved:
> >
> > 1. The work remains heavily engineered around the FLUX. Adding a single model to the Appendix feels more like a patch than a demonstration of a truly methodological advancement.
> >
> > 2. As acknowledged, the dataset skews significantly towards graphic design/cartoon styles. Relying on "future plans" to address the lack of photorealism limits the current scientific value of the work.
> >
> > 3. The adjustments described in the rebuttal focus on captions rather than improving actual figure readability.
> >
> > The submission reads more like a technical report or resource release rather than a fundamental research paper with the novelty expected at ICLR. Therefore, I tend to maintain my score.

---

> > > ### Author Response · Authors · 2025-11-28
> > >
> > > We thank the reviewer for the follow-up comments. We respectfully clarify several key points regarding the contribution, methodological substance, and value of our work.
> > >
> > > ------
> > >
> > > ## 1. Dataset/benchmark track papers are evaluated on methodological rigor and community value—not solely algorithmic novelty
> > >
> > > > High-quality open datasets have become a critical research infrastructure. Top AI conferences such as ICLR and NeurIPS explicitly encourage dataset-and-benchmark papers. We emphasize that for **dataset and benchmark track papers**, the focus should be on **method effectiveness, task difficulty, data quality, and contribution to the open-source community**, rather than solely on proposing new algorithms or model architectures.
> > >
> > > > Our method constructs a systematic, automatic data generation and filtering pipeline based on existing open-source pre-trained models. While it utilizes the capability of the pre-trained FLUX model, **our novel framework enables FLUX to generate multi-layer images with style consistency, a capability it previously lacked.**
> > >
> > > > This systematic integration of existing agents to achieve significant performance gains, similar to works like Cursor [3], foundational research in Agentic AI (ReAct, Tree of Thoughts, Cognitive Architectures for Language Agents) [1, 2, 4], and advanced deep research systems (Assisting in Writing Wikipedia-like Articles, and the Deep Research System Card) [5, 6], holds substantial value for the community.
> > >
> > > > Regarding generalizability, we have demonstrated its successful migration and performance using Qwen-Image instead of FLUX in the appendix, and we are adding more model results.
> > >
> > > ## 2. Our pipeline is not FLUX-specific; it adds capabilities that the base model does not possess
> > >
> > > > While FLUX.1-[dev] is a strong base model, **it cannot generate coherent, editable multi-layer images on its own**. Our contribution is a model-agnostic, multi-stage pipeline that adds new capabilities through data construction, layer-aware prompting, matting, and structural filtering.
> > >
> > > > We already demonstrated generalization on Qwen-Image in the appendix and are adding more models in the revised version.
> > >
> > > ## 3. This work aims to release the first high-quality, open-source multi-layer design image dataset to the community, marking a pioneering "from 0 to 1" breakthrough in this exceptionally challenging field.
> > >
> > > > Multi-layer design generation and multi-layer real image generation **are both challenging and equally vital fields**, where the design generation component enables crucial applications like fine-tunable posters. Prior work (ART [7]) demonstrated that models trained on high-quality design data generalize effectively, thus making a strong multi-layer design dataset foundational for real-world data creation. Having overcome significant hurdles by contributing our proposed first large-scale dataset, we detail the inherent difficulties of this field below.
> > >
> > > ### (1) Obtaining large-scale, semantically meaningful multi-layer layouts
> > >
> > > > - Commercial design data contains heterogeneous structures, inconsistent layer roles, and no textual supervision.
> > > >
> > > > - We collect 800K real multi-layer designs and generate **layer-wise captions + global composition descriptions** using Llava 1.6 and GPT-4o, preserving the semantic structure and layer ordering.
> > >
> > > ### (2) Preventing cross-layer conflicts in generated multi-layer images
> > >
> > > > - Naïve independent layer generation leads to duplicated objects, spatial conflicts, and inconsistent transparency.
> > > >
> > > > - We manually annotate 8K artifact cases and fine-tune an **Artifact Classifier**—the first structural validator for multi-layer images—to filter 200K reliable samples.
> > >
> > > ### (3) Ensuring transparent layer quality at scale
> > >
> > > > - Clean alpha mattes are extremely hard to evaluate automatically.
> > > >
> > > > - We propose **TIPS (Transparent Image Preference Score)**, trained on curated transparent-layer data from MultilayerTrain, LayerDiffuse, and our reproduced LayerDiffuse-FLUX, enabling fine-grained layer-level quality control.
> > >
> > > ### (4) Preserving semantic structure during style-conditioned multi-layer generation
> > >
> > > > - We define 21 design styles and use GPT-4o to rewrite style-directed prompts, while enforcing structural consistency through combined TIPS + Artifact Classifier filtering.
> > >
> > > ### (5) Preventing aesthetic drift during large-scale generation
> > >
> > > > - Fine-tuning strong models on noisy or structurally unstable multilayer data usually harms aesthetic performance.
> > > >
> > > > - Our two-stage pipeline (PrismLayers → PrismLayersPro) maintains FLUX’s original quality (as shown in Fig. 12), while naive alternatives significantly degrade aesthetics.
> > > >
> > > > - Although not our primary goal, we have also supplemented our paper with 1k real-world multi-layer images generated by our method, demonstrating its generalizability and marking our essential step forward in the multi-layer dataset domain.

---

> > > > ### Author Response · Authors · 2025-11-28
> > > >
> > > > ## 4. The dataset is already enabling subsequent research
> > > >
> > > > > - We also want to emphasize that the introduction of our dataset has attracted increasing research attention to the multi-layer design generation field.
> > > > >
> > > > > - Our dataset is currently the **only high-quality open-source resource** for multi-layer generation and has already been adopted by follow-up work in the community.
> > > > > This demonstrates concrete scientific impact.
> > > >
> > > > Finally, we have further improved the writing and readability of the figures based on the reviewers' feedback in the revised paper.
> > > >
> > > > ## References
> > > >
> > > > > [1] Yao, S., Zhao, J., Yu, D., Du, N., Shafran, I., Narasimhan, K., & Cao, Y. (2022). ReAct: Synergizing Reasoning and Acting in Language Models. arXiv preprint arXiv:2210.03629.
> > > > >
> > > > > [2] Yao, S., Yu, D., Zhao, J., Shafran, I., Griffiths, T. L., Cao, Y., & Narasimhan, K. (2023). Tree of Thoughts: Deliberate Problem Solving with Large Language Models. Advances in Neural Information Processing Systems 36 (NeurIPS 2023).
> > > > >
> > > > > [3] Cursor. Retrieved from https://cursor.sh/
> > > > >
> > > > > [4] Yao, S., Sumers, T., Narasimhan, K., & Griffiths, T. L. (2023). Cognitive Architectures for Language Agents. Transactions on Machine Learning Research (TMLR 2024).
> > > > >
> > > > > [5] Shao, Y., Jiang, Y., Kanell, T. A., Xu, P., Khattab, O., & Lam, M. S. (2024). Assisting in Writing Wikipedia-like Articles From Scratch with Large Language Models. arXiv preprint arXiv:2402.14207.
> > > > >
> > > > > [6] OpenAI. (2025). Deep Research System Card. OpenAI Technical Report. Retrieved from https://cdn.openai.com/deep-research-system-card.pdf
> > > > >
> > > > > [7] Pu, Y., Zhao, Y., Tang, Z., et al. (2025). ART: Anonymous region transformer for variable multi-layer transparent image generation. Proceedings of the Computer Vision and Pattern Recognition Conference 2025 (CVPR 2025).

---

### Official Review · Reviewer_vNNp · 2025-10-30

**Soundness:** 2
**Presentation:** 2
**Contribution:** 2
**Rating:** 4
**Confidence:** 3

**Summary:**

The paper proposes a training-free synthesis pipeline for high-quality multi-layer transparent images: LayerFLUX adopts a "generate-then-matting" scheme, using suffix prompts to guide diffusion models in generating single-layer images with uniform backgrounds and leveraging RMBG-2.0 for accurate alpha matte extraction, while MultiLayerFLUX composes these layers by preserving their original aspect ratios and following semantic layouts. The paper curates 4 open high-fidelity datasets—PrismLayers (200K), PrismLayersPlus (100K), PrismLayersPro (20K), and PrismLayersReal (1K)—curated via artifact filtering, TIPS evaluation (a CLIP-fine-tuned transparent image quality model), and human selection (for PrismLayersPro). For evaluation, the paper fine-tunes the ART model on PrismLayersPro to obtain ART+, which outperforms the original ART of head-to-head user study comparisons and approaches the visual quality of FLUX.1-[dev]. Additionally, LayerFLUX outperforms LayerDiffuse on Layer-Bench with 63.1% win rate in layer quality and 61.2% in prompt following. The TIPS model provides a dedicated metric to assess the aesthetic quality of transparent images, validating the effectiveness of the proposed methods and datasets.

**Strengths:**

- High-quality and comprehensive open-source datasets. The paper constructs four Multi-Layer Transparent Image datasets, with the number of layers ranging up to 50. This makes substantial contributions to the field's advancement.
- Pipeline for constructing high-quality multilayer transparent datasets. The paper proposes a novel pipeline that leverages FLUX—a powerful full-image generation model—to produce multilayer transparent images.
- A new preference model for transparent image synthesis. Addressing the incompatibility of existing RGB image quality models with transparent layers, the paper introduces the TIPS (Transparent Image Preference Score) model, which fills a current research gap.

**Weaknesses:**

- Lack of naturalness in fully synthesized datasets. Visualizations indicate the proposed dataset contains numerous cartoon elements that are generally easy to matte out from images. However, the paper does not propose a new approach to acquire transparent images from real-world scenes.

- Failure to address key challenges in multi-layered image generation. For multi-layered image generation, shadows, lighting, transparent objects (e.g., glass), and reflections on water or other surfaces are critical. When adding an object to a scene, the image should change in line with physical laws—a key requirement that the proposed method does not fulfill.

**Questions:**

- When constructing multi-layered image datasets, how does the pipeline define the semantic content of each layer while explicitly modeling physical interactions (e.g., shadow casting, light propagation, and material properties like transparency/reflectivity) across layers?
-  How does the proposed pipeline ensure visual coherence and physical plausibility in generated multi-layered images?
- Are the data construction pipeline and benchmark suite (including pre-trained models, evaluation protocols) planned for open-source release?

---

> ### Author Response · Authors · 2025-11-23
>
> Thank you for your careful review and for recognizing the contributions of our datasets, pipeline, and TIPS model. Below, we first summarize our understanding of your comments and then respond to each concern.
>
> **Strengths you highlighted**
> - Four high-quality, comprehensive open-source datasets (PrismLayers / Plus / Pro / Real) with up to 50 layers per image.
> - A pipeline that leverages FLUX to construct high-quality multi-layer transparent datasets.
> - A new Transparent Image Preference Score (TIPS) model that fills a gap where existing RGB-oriented aesthetic models do not directly apply to transparent layers.
>
> **Main concerns / weaknesses**
> 1. The datasets remain heavily synthetic and lack natural complexity, especially for transparent or physically challenging real-world scenes.
> 2. The method does not address essential physical interactions (shadows, lighting, transparency, reflections) required for physically plausible multi-layer compositing.
> 3. How do we ensure visual coherence and physical plausibility of generated multi-layer images?
> 4. Are the data construction pipeline and benchmark (pretrained models, evaluation protocols) planned to be open-sourced?
>
> **Naturalness and real-world transparent data**
>
> We agree that our datasets contain many stylized / cartoon-like elements and that this reduces naturalness in the strictly photographic sense. This is, however, aligned with our target domain:
> - Graphic design, UI/UX, AR assets, and stickers are inherently stylized and often created in vector-like or semi-cartoon styles.
> - For these tasks, the ability to edit layers independently (text, icons, characters, background patterns) is more important than matching real-world lighting.
> Among existing datasets:
> - MLTD (Pu et al., 2025) also focuses on graphic design, is not open-sourced, and has less consistent alpha quality.
> - MuLAn (Tudosiu et al., 2024) and MLCID (Huang et al., 2024) collect multi-layer data from COCO/LAION but have poor aesthetics and alpha quality (Table 1).
> - None provides an open, large-scale, high-fidelity basis for layered editing comparable to PrismLayersPro.
> In Appendix N, we describe our top-down pipeline, which can be further extended to real-world photos. We are currently extending this pipeline to larger real-image pools (including non-generated photos), and plan to release a real split in future work.
>
> **Physical interactions (shadows, lighting, transparency, reflections)**
>
> Your second weakness points to an important and largely open challenge in multi-layer generation: physically accurate compositing.
> Our focus in this paper is on:
> - semantic and perceptual coherence of layered compositions, and
> - providing high-quality, editable RGBA assets that can enable future physically aware models.
> As far as we know, existing multi-layer generative methods such as LayerDiff (Huang et al., 2024) and ART (Pu et al., 2025) also do not explicitly simulate these physical processes; they treat layers as learned RGBA tokens and rely on the model to implicitly infer plausible interactions from data.
> Our datasets and pipeline can serve as a strong training and evaluation resource for such models, including future ones that incorporate explicit physics.

---

> > ### Author Response · Authors · 2025-11-23
> >
> > **Answers to your specific questions**
> >
> > **(1) Semantic content and physical interactions per layer**
> >
> > Each layer in our 800K internal dataset comes with layer ordering and semantic roles (background, character, text, decorations, icons, etc.). We extract:
> > - global and layer-wise captions using LLaVA 1.6 and GPT-4o, and
> > - a semantic layout consisting of bounding boxes and z-order.
> > In MultiLayerFLUX, we do not explicitly model physical interactions such as shadows. Instead, we:
> > - treat each layer’s caption and box as defining its semantic content and position,
> > - ensure stylistic and semantic consistency via shared global prompts and seeds, and
> > - rely on artifact filtering + human selection (for PrismLayersPro) to remove obviously incoherent compositions.
> >
> > **(2) Visual coherence and physical plausibility**
> >
> > Visual coherence is enforced through several mechanisms:
> > - global consistency: all layers for a sample share a single global scene prompt and seed, producing consistent style and color.
> > - Layout consistency: bounding boxes and z-orders are inherited from real designs; we avoid excessive overlap via heuristics and an Artifact Classifier trained on examples of bad overlap or duplication (Figure 7).
> > - Data-driven coherence: fine-tuning ART on PrismLayersPro (ART+) substantially improves global harmonization compared to both ART and MultiLayerFLUX, as confirmed by user studies (Figure 2).
> > Physical plausibility (shadows, reflections) is not fully guaranteed; we will explicitly acknowledge this limitation and outline the extensions described above.
> >
> > **(3) Open-sourcing pipeline, models, and benchmark**
> >
> > Yes — we have already uploaded PrismLayers / Plus / Pro / Real to HuggingFace under an anonymous account for the review process.
> > In addition, we plan to open-source our model components and evaluation resources, including
> > - the LayerFLUX / MultiLayerFLUX scripts,
> > - the TIPS model and our artifact classifier,
> > - and the evaluation metrics and protocols.
> >
> > We appreciate your thoughtful comments and will integrate your suggestions in the revision, and we hope that our detailed clarifications help convey the strengths of our work and lead to a more favorable final evaluation.

---

### Official Review · Reviewer_7bLz · 2025-11-01

**Soundness:** 2
**Presentation:** 3
**Contribution:** 2
**Rating:** 4
**Confidence:** 4

**Summary:**

The authors propose a new series of datasets for multi-layer generation, where the proposed dataset mainly focuses on artistic elements from the qualitative examples present in the paper. The proposed dataset has multiple versions which are differing in terms of data scale and the types of examples included. To generate such data, the authors propose an inference time methodology named LayerFLUX and MultiLayerFLUX, which is integrated on a multi-stage data generation pipeline. The proposed dataset includes samples that consists of many layers compared to existing layered datasets, which is an advantage of the method. Lastly, the authors show the quality of the collected data with improvements reported on ART+ model.

**Strengths:**

- The authors propose a series of datasets that include a significant number of layers and in different scales with changing quality. Unlike the existing datasets, the proposed dataset shows a data scenario that can be generalized better to layered synthesis in imaging scenarios.
- In addition to layers, the proposed dataset also gives a good source of text rendering data.
- The paper proposes a multi-stage pipeline to generate a high quality data generation. The proposed pipeline covers the quality issues and the filtering of data in a sensible way.

**Weaknesses:**

- The related work section is limited in the paper and do not cover the majority of layered synthesis work. While it is understandable to not include in the main paper, such works should be acknowledged at least in the supplementary.
- From the data samples included in the dataset, even in the Real data split of the proposed data, the samples seem artificial and do not serve as real samples.
- As a primary use case of layered images, objects involving transparency properties have crucial importance. While samples including such objects are shown qualitatively, dataset statistics on such objects would be helpful to assess the usability of the proposed dataset.
- While showcasing a dataset with considerably higher quality, the contribution made with LayerFLUX and MultiLayerFLUX is questionable. From the text, it seems that the layered generation is only initiated with suffix prompts fed into FLUX. In addition, the paper states that MultiiLayerFLUX is generating multiple single layer images with varying resolutions. After applying the matting stage, one question arises is that how does these layers belong to the same scene (enforced by prompting, or an attention sharing mechanism). Explaining such mechanics is important to be able to understand how can multi-layer synthesis can be facilitated. In the current format the operation "Compose" is very ambiguous and can best be interpreted as alpha blending.
- The organization of the paper can be improved, the current organization is fairly hard to follow. The figures can be better connected with the text with more explanatory captions.

**Questions:**

- How did the authors assess the quality of the data in Table 1 (Alpha quality, aesthetic)?
- In the multi-layer generation pipeline with MultiLayerFLUX, how does the approach handle overlapping layers, and render them in a coherent scene?
- In the examples provided for ART and ART+, the difference is not that clear. What is the main point that the readers should pay attention there? To be able to understand the quality difference (if any), we should be provided with generation prompts.

---

> ### Author Response · Authors · 2025-11-23
>
> Thank you for your thoughtful and balanced review. We appreciate your recognition of the dataset design and pipeline, and we address your concerns point by point below.
>
> **Strengths you highlighted**
> - A series of datasets with many layers and multiple scales, which can better generalize to layered synthesis scenarios than previous resources.
> - The datasets provide a valuable source for text rendering.
> - The proposed multi-stage pipeline (LayerFLUX + MultiLayerFLUX + filtering) sensibly handles quality issues.
>
> **Main weaknesses**
> 1. The related work section is limited and does not fully cover layered synthesis works.
> 2. Even the "Real" subset appears somewhat artificial and not truly real.
> 3. For transparent objects (glass, water, etc.), statistics would help assess the dataset’s usability.
> 4. The contribution of LayerFLUX and MultiLayerFLUX is unclear: they appear to just use suffix prompts with FLUX and then alpha blending, and it is ambiguous how layers are enforced to belong to the same scene.
> 5. Paper organization and figure-caption connections could be improved.
>
> **On related work coverage**
>
> Thank you for your feedback. We agree that our previous Related Works section was overly concise due to space limitations. In the revised version of the paper, we have positively acknowledged and elaborated on the relevant work concerning Transparent Layer Generation and Graphic Design Generation. The modifications have been highlighted in blue.
>
> **Real subset and realism**
>
> Your observation that even the Real split looks somewhat artificial is fair. Our current Real subset (1K examples) is generated using a top-down scheme (Appendix N) starting from FLUX-generated photos, followed by detection, matting, inpainting, and manual selection. As such, it is closer to stylized photography than raw, in-the-wild photographs.
> We acknowledge that truly real-world multi-layer images with high-quality alpha mattes are extremely challenging to obtain at scale, due to the difficulty of collecting layered images and annotating precise alpha mattes.
> In future work, we plan to explore semi-automatic pipelines that start from in-the-wild layered images and improve our pipeline to obtain higher-quality real multi-layer datasets.
>
> **Statistics for transparent objects**
>
> We agree that objects with intrinsic transparency (glass, liquids, water surfaces, etc.) are particularly important for layered images.
> Our current paper qualitatively shows such examples (e.g., Figure 6 and 13), but we did not provide dedicated statistics. In the revision, we will: (i) add quantitative statistics on transparent object categories in the datasets, and (ii) highlight more examples of such objects in the supplementary material.
>
> **Clarifying the novelty and mechanics of LayerFLUX / MultiLayerFLUX**
>
> We appreciate your request to better separate the contributions of LayerFLUX and MultiLayerFLUX from "just suffix prompts + matting + alpha blending".
> Compared to MAGICK (Burgert et al., 2024) and LayerDiffuse (Zhang & Agrawala, 2024), LayerFLUX:
> - is training-free and model-agnostic (no finetuning of FLUX needed),
> - uses carefully studied suffix prompts such as "isolated on a solid gray background" to induce backgrounds that are easy to matte, with quantitative attention-map analysis and matting quality comparison across suffix variants (Tables 9–11, Appendix L–M),
> - evaluates multiple matting backbones (SAM2, BiRefNet, RMBG-2.0) and selects RMBG-2.0 as best on Layer-Bench, and
> - empirically outperforms LayerDiffuse on user studies across three prompt types on Layer-Bench (in Figure 15, win rates of 63.1% and 61.2% for layer quality and prompt following).
> **MultiLayerFLUX** further extends LayerFLUX to multi-layer images based on semantic layouts extracted from 800K real-world designs:
> - All layers for a sample share a global scene prompt, promoting a consistent style and color palette.
> - Each layer is associated with a semantic role (e.g., background pattern, main character, text block, icon), a bounding box, and a z-order derived from the original crawled design.
> - Instead of generating every layer in a fixed square canvas, we preserve the aspect ratio, generate at variable resolutions (long side 1024), and then resize into the assigned bounding box before compositing.
> Thus, MultiLayerFLUX is not just “arbitrary alpha blending”; it is layout-conditioned, role-aware composition based on real design layouts.
> In comparison, LayerDiff (Huang et al., 2024) and ART (Pu et al., 2025) use top-down, jointly trained diffusion architectures to model multiple layers, whereas MultiLayerFLUX provides a bottom-up, training-free alternative that can immediately bootstrap large datasets from any strong full-image generator.

---

> > ### Author Response · Authors · 2025-11-23
> >
> > **Paper organization and figure captions**
> >
> > We acknowledge that the current organization of the paper's Methodology and Experiments sections may be difficult to follow due to the large amount of content. We greatly appreciate your suggestion and are happy to outline our structure for you. In the Methodology section, we primarily introduce the data implementation pipeline, organized along the lines of statistical information, construction process, and specific module methods, and in the Experiments section, we mainly cover the training setup for the $\text{ART}+$ model, as well as the qualitative and quantitative evaluation results. The overall organizational structure should not be overly complex; we will ensure a thorough revision and refinement in the final version to provide a clearer narration and ensure a smoother flow.
> > In addition, we have revised the captions for the figures and tables in the article (with modified parts highlighted in blue), hoping this will help you better understand the content.
> >
> > **Answering your specific questions**
> >
> > **(a) Quality assessment in Table 1 (Alpha quality, Aesthetic)**
> >
> > We apologize for not making this explicit.
> > - For our datasets (PrismLayers / Plus / Pro / Real), we compute the mean TIPS score over all layers, then bucketize the distributions into "poor / normal / good / excellent" based on quantile thresholds learned from human preference data. PrismLayersPro achieves the highest TIPS distribution, hence "excellent".
> > - For alpha quality, we:
> >   - sample a subset of images from each dataset,
> >   - inspect alpha boundaries and semi-transparent regions,
> >   - cross-check with matting outputs (where available), and
> >   - assign categorical ratings ("poor / normal / good") based on edge sharpness, hole artifacts, and background leakage.
> > - For external datasets (MuLAn, MLTD, etc.), we follow the same sampling and human evaluation protocol, which leads to the qualitative ratings reported in Table 1.
> >
> > **(b) Overlapping layers and scene coherence in MultiLayerFLUX**
> >
> > As noted above, MultiLayerFLUX uses semantic layouts from the crawled 800K designs, which include bounding boxes and z-orders. During composition:
> > - We preserve each layer's aspect ratio and place it into its assigned bounding box.
> > - We train an Artifact Classifier (based on BLIP-2) to detect and filter out samples with duplicated layers in conflicting positions or visually unreasonable overlap patterns (Figure 7).
> > This combination of layout constraints and artifact filtering enforces semantic coherence layers from one design.
> >
> > **(c) ART vs. ART+ examples: differences and prompts**
> >
> > We agree that the current figure does not clearly communicate where ART+ is better.
> > In the revision, we provide the exact prompts used in each comparison and add zoomed-in crops highlighting.
> > The key improvements are:
> > - Layer quality: ART+ reduces noisy edges and incomplete matting, as also reflected by improved TIPS and FID merged on FLUX-Multi-Layer-Bench.
> > - Prompt following: text and small decorative details more closely match the prompts.
> > - Global harmonization: layers fit together more consistently than in ART (Figure 10–12).
> >
> > We look forward to engaging further in the discussion and sincerely hope our clarifications address your concerns and merit a more positive reassessment. Thank you again for your constructive feedback.

---

> ### Comment · Reviewer_7bLz · 2025-11-28
> **Thank you for your response**
>
> I thank the authors for their timely clarifications and their adjustments to the paper. Over the presented statistics and extended sections, I believe that it would also be beneficial to demonstrate the qualitative performance on objects with intrinsic transparency properties and if that is a failure case, it can be stated more clearly (Figure 14 column 3 and Figure 15 row 1 made me question this property a bit). Also I think it would be helpful to provide some examples from example semantic layouts (like in form of bounding boxes etc.) to show whether the provided layouts include any form of occlusions, and if it is it would be a strength of the dataset. If the authors can add such adjustments, I am willing to increase my score.

---

> ### Author Response · Authors · 2025-12-03
> **Thank you for your willingness to re-evaluate our work**
>
> Thank you very much for your prompt response and for considering raising your score. We greatly appreciate your constructive suggestions regarding intrinsic transparency and occlusion layouts.
>
> In response to your specific requests, we have updated our paper and supplementary material with **Appendix Q** and **Appendix R**. Here is a summary of the new evidence and analyses:
>
> 1. **Analysis on Objects with Intrinsic Transparency (Appendix Q)** As suggested, we conducted a dedicated quantitative and qualitative evaluation on 25 categories of transparent objects (e.g., glass, ice, water) using 100 prompts generated by Qwen3-VL [1].
>   - **Quantitative**: We evaluated the alpha quality using TIPS scores. Results show that RMBG-2.0 achieves an average TIPS score of 26.78, outperforming SAM 3 [2] (24.15).
>   - **Failure Cases & Limitations**: We have explicitly discussed the limitations. We agree that recent segmentation methods still struggle to recover fine-grained alpha values for complex semi-transparent regions. We have categorized this as an open challenge in the field. All generated samples and prompts are provided in the supplementary material for inspection.
> 2. **Overlapping Layers and Semantic Layouts (Appendix R)** To demonstrate the occlusion properties of our dataset, we added statistics and visual examples in Appendix R:
>   - **Statistics**: Our analysis reveals that PrismLayersPro contains an average of 6.32 overlapping layer pairs per image, confirming that our layouts feature rich occlusion patterns rather than simple non-overlapping placements.
>   - **Physical Plausibility**: We clarified that the z-order of these layers is derived from real-world design data and further refined by our Artifact Detector to ensure that occlusions are physically and visually plausible (e.g., background patterns behind characters).
> We hope these additional experiments and clarifications satisfactorily address your remaining concerns regarding transparency failure cases and layout occlusions. We sincerely thank you again for your time and for your willingness to re-evaluate our work.
>
> > [1] Bai, Shuai, et al. "Qwen3-VL Technical Report." arXiv preprint arXiv:2511.21631 (2025).
> >
> > [2] Carion, Nicolas, et al. "SAM 3: Segment Anything with Concepts." arXiv preprint arXiv:2511.16719 (2025).

---

### Official Review · Reviewer_uqof · 2025-11-01

**Soundness:** 3
**Presentation:** 3
**Contribution:** 3
**Rating:** 6
**Confidence:** 4

**Summary:**

This paper introduces PrismLayers, a suite of four open, high-quality datasets (PrismLayers, PrismLayersPlus, PrismLayersPro, and PrismLayersReal) for multi-layer transparent image generation, addressing the lack of large, high-fidelity data in this domain. The authors present a training-free synthesis pipeline using off-the-shelf diffusion models and advanced matting techniques to generate and curate these datasets, which include accurate alpha mattes and diverse styles. They also propose LayerFLUX for single-layer generation and MultiLayerFLUX for compositing, and demonstrate that fine-tuning the ART model on PrismLayersPro yields ART+, a model that outperforms previous methods in user studies and matches the visual quality of leading text-to-image models. The work establishes a foundation for research and applications requiring editable, visually compelling, and precisely layered image geneartions.

**Strengths:**

+ Releases the first high-fidelity datasets for multi-layer transparent image generation, filling a major gap in the field.

+ The work employs rigorous artifact filtering, aesthetic scoring, and human selection to ensure dataset quality and diversity. The creating pipline and experiences are also valuable.

+ Fine-tuned models (ART+) trained on PrismLayersPro achieve better performance in both quantitative metrics and user studies compared with recent single-layer models.

**Weaknesses:**

- While high-quality, the datasets are primarily synthetic, and may not fully capture the complexity or coherence of real-world multi-layer images.

- The accuracy of text rendering seems one of the eval dimension for this work. But it seems that in both user studies and metrics. How about the text rendering quality/accuracy of the proposed work?

**Questions:**

Please refer to the detailed questions raised in Weakness section above.

---

> ### Author Response · Authors · 2025-11-23
>
> We sincerely thank you for your detailed and constructive review, as well as for your positive assessment of the overall contribution (Score 6, all “good” on soundness, presentation, and contribution). Below, we first summarize our understanding of your comments and then respond to each concern.
>
> **Strengths you highlighted**
> - We provide the first family of open, high-fidelity multi-layer transparent image datasets (PrismLayers / Plus / Pro / Real), with accurate alpha mattes, diverse styles, and layer-wise annotations.
> - The training-free data generation pipeline (LayerFLUX + MultiLayerFLUX), together with rigorous filtering via the Artifact Classifier and TIPS plus human selection, is well-designed and practically useful.
> - Fine-tuning ART on PrismLayersPro yields ART+, which is preferred in user studies and approaches the visual quality of strong text-to-image models.
>
> **Main concerns / weaknesses**
> - Although the datasets are high-quality, they are predominantly synthetic and may not fully capture the complexity and coherence of real-world multi-layer images.
> - Since text rendering is an important part of the task, you ask how we evaluate the text rendering quality/accuracy of our methods, beyond the global metrics and user studies currently reported.
>
> **On synthetic vs. real multi-layer data**
> We agree that relying mainly on synthetic data is a limitation. This is an explicit trade-off we make in order to achieve:
> - precise alpha mattes and layer masks at scale,
> - rich layer counts (up to 50) and text-centric layers, and
> - strong control over prompts and layouts (global + layer-wise captions).
> One of our target applications is graphic design, UI/UX assets, and compositing-ready elements, where content is typically designed, stylized, or semi-cartoonish rather than purely photographic. In this domain, we are not aware of any prior open multi-layer dataset with comparable scale, alpha fidelity, and aesthetics:
> - MuLAn (Tudosiu et al., 2024) provides 44K multi-layer images with 2–6 layers, but is mostly based on COCO/LAION and features relatively low alpha and aesthetic quality.
> - MLTD (Pu et al., 2025) contains ∼1M design-style multi-layer images with up to 50 layers, but is not open-sourced and has more limited alpha and aesthetic quality (Table 1).
> - MLCID (Huang et al., 2024) focuses on 2–4 layers with relatively poor alpha and aesthetics.
> In contrast, PrismLayersPro is, to our knowledge, the first open dataset that simultaneously offers: (i) large scale, (ii) 2–50 layers, (iii) high-quality alpha mattes obtained via state-of-the-art matting (RMBG-2.0), and (iv) excellent aesthetic quality as measured by our TIPS model and human selection.
> We acknowledge that PrismLayersReal (1K) is small and cannot fully represent real-world photo composites. This is why we position it as a bridging set rather than a main training resource. In Appendix N, we already describe a top-down real-image pipeline that starts from FLUX-generated photos, applies object detection and matting, and then manual selection to obtain high-quality photorealistic layers.
> We would greatly appreciate it if you could let us know whether our clarifications on the synthetic–real trade-off and dataset scope sufficiently address your concerns.
>
> **Revision**
> In the revised version, we clarify the design-centric scope of our datasets in the introduction and discussion, explicitly stating that we target graphic design and layered editing workflows rather than physically faithful photographic scenes.
>
> **On text rendering quality and evaluation**
> We appreciate your emphasis on text rendering, which is indeed central to many designs (logos, posters, banners, etc.).
> In Section 3.1 quantitatively analyzes the number of text layers per image, characters per instance, and area ratios of text layers, highlighting that PrismLayers intentionally contains a large amount of well-isolated visual text.
> Moreover:
> - In Figure 2, our user studies ask participants to rate prompt following and layer quality; illegible or misspelled text visibly reduces preference.
> - Our TIPS model, trained on human preference data for transparent images, implicitly penalizes noisy or unreadable text layers.
> However, we agree that these are indirect and that a dedicated text-rendering evaluation would strengthen the paper.
>
> **Planned text-rendering evaluation**
> We are currently extracting and processing the relevant generated samples for detailed text-rendering statistics, and we aim to provide an OCR-based metric that addresses your concerns.
>
> We will actively incorporate your suggestions during the discussion period, including adding the text-rendering evaluation, and we sincerely hope our clarifications address your concerns and merit a more positive reassessment. Thank you again for your constructive feedback.

---

### Author Response · Authors · 2025-11-23

We have carefully addressed all concerns, and all major modifications and new results are highlighted in blue in the revised manuscript.

In particular, to address common concerns on real-image realism, physical plausibility across layers, and FLUX-centric evaluation, we (i) extend our multi-layer engine to Qwen-Image to demonstrate the model-agnostic nature of our training-free pipeline (Appendix O), and (ii) build on the original photorealistic engine in Appendix N to propose an advanced photorealistic multi-layer data engine using Qwen-Image-Edit that alleviates occlusion and improves cross-layer consistency (Appendix P). We are currently using this upgraded engine to further improve the quality and scale of PrismLayersReal.

---

### Author Response · Authors · 2025-12-04
**Response to All Reviewers and AC**

### **3. Clarification on Real-World Data & Technical Challenges *(reviewers uqof, vNNp, 7bLz, H3ue)***

Regarding the 1K real-world subset, we clarify that it serves to demonstrate generalizability, while our primary contribution is the ***first large-scale, high-quality dataset for multi-layer design generation***. We emphasize that design generation is an equally vital field, enabling applications like editable posters. Crucially, ***prior work (ART [7]) demonstrated that models trained on high-quality design data generalize effectively to real-world multi-layer generation, establishing our dataset as a foundational resource for the broader field***.

To construct this dataset, we overcame five specific technical hurdles:
- **Layout Extraction**: Mining 800K designs and generating precise captions via LLaVA 1.6 and GPT-4o.
- **Conflict Prevention**: Developing the first Artifact Classifier (trained on 8K annotations) to filter cross-layer conflicts.
- **Quality Control**: Ensuring transparent layer quality at scale via our TIPS metric.
- **Structural Consistency**: Enforcing 21 defined styles and semantic structures using a combined TIPS and Artifact Classifier filter.
- **Aesthetic Preservation**: Preventing drift via a two-stage pipeline (PrismLayers → PrismLayersPro) to preserve the original text-to-image generation capabilities during fine-tuning.

Beyond the specific technical clarifications discussed above, we have comprehensively updated the manuscript to incorporate ***all requested experiments, improve the presentation, and refine the methodology***. Notably, ***Reviewer 7bLz explicitly stated an intent to raise their score***. While the discussion period concluded before formal updates were possible for the remaining reviewers, we are confident that our detailed responses have fully resolved their concerns and clarified all misunderstandings.

# References

> [1] Yao, S., Zhao, J., Yu, D., Du, N., Shafran, I., Narasimhan, K., & Cao, Y. (2022). ReAct: Synergizing Reasoning and Acting in Language Models. arXiv preprint arXiv:2210.03629.
>
> [2] Yao, S., Yu, D., Zhao, J., Shafran, I., Griffiths, T. L., Cao, Y., & Narasimhan, K. (2023). Tree of Thoughts: Deliberate Problem Solving with Large Language Models. Advances in Neural Information Processing Systems 36 (NeurIPS 2023).
>
> [3] Cursor. Retrieved from https://cursor.sh/
>
> [4] Yao, S., Sumers, T., Narasimhan, K., & Griffiths, T. L. (2023). Cognitive Architectures for Language Agents. Transactions on Machine Learning Research (TMLR 2024).
>
> [5] Shao, Y., Jiang, Y., Kanell, T. A., Xu, P., Khattab, O., & Lam, M. S. (2024). Assisting in Writing Wikipedia-like Articles From Scratch with Large Language Models. arXiv preprint arXiv:2402.14207.
>
> [6] OpenAI. (2025). Deep Research System Card. OpenAI Technical Report. Retrieved from https://cdn.openai.com/deep-research-system-card.pdf
>
> [7] Pu, Y., Zhao, Y., Tang, Z., et al. (2025). ART: Anonymous region transformer for variable multi-layer transparent image generation. Proceedings of the Computer Vision and Pattern Recognition Conference 2025 (CVPR 2025).
>
> [8] Zhang, L., & Agrawala, M. (2024). Transparent Image Layer Diffusion using Latent Transparency. ACM SIGGRAPH 2024 Conference Proceedings.
>
> [9] Huang, R., Cai, K., Han, J., Liang, X., Pei, R., Lu, G., Xu, S., Zhang, W., & Xu, H. (2024). LayerDiff: Exploring Text-guided Multi-layered Composable Image Synthesis via Layer-Collaborative Diffusion Model. arXiv preprint arXiv:2403.11929.
>
> [10] Wu, C., Li, J., Zhou, J., Lin, J., Gao, K., Yan, K., Yin, S., Bai, S., Xu, X., Chen, Y., et al. (2025). Qwen-Image Technical Report. arXiv preprint arXiv:2508.02324.

---

### Author Response · Authors · 2025-12-04
**Response to All Reviewers and AC**

Dear reviewers and AC,

We appreciate your time in reviewing our paper and rebuttal. Throughout the discussion phase, we received valuable comments and noted a clear intention from reviewers to raise their scores. It is regrettable that these updates were not formally captured in the system. To ensure our work is accurately assessed, we would like to briefly recap our main contributions and address the remaining concerns and misunderstandings below.

## **Contribution**

This work presents ***the first large-scale, high-quality dataset for multi-layer design imagery***, marking a pioneering breakthrough in this challenging domain. To address the fundamental data scarcity, we release a suite of four open, ultra-high-fidelity datasets (200K images with accurate alpha mattes) alongside a novel training-free synthesis pipeline. Furthermore, we introduce ART+, a state-of-the-art multi-layer generation model. Fine-tuned on our data, ART+ significantly outperforms prior arts and matches the aesthetic quality of modern text-to-image models. ***Our work establishes a critical foundation, enabling future research and applications requiring precise, editable, and visually compelling layered imagery.***

## **Concerns during rebuttal**

### **1. Methodological Innovation: Agentic Integration as a Core Contribution *(reviewer H3ue)***

- **Prioritizing Methodological Rigor**: High-quality open datasets act as critical research infrastructure. We emphasize that for Dataset and Benchmark tracks, ***evaluation should prioritize methodological rigor, data quality, and community contribution, rather than solely on the novelty of underlying model architectures.***
- **Systematic Integration via Agentic AI**: Our contribution lies in constructing a systematic, automated generation pipeline. While we leverage pre-trained models like FLUX, ***our novel framework unlocks a critical capability they previously lacked: the generation of style-consistent, multi-layer images***. This approach—systematically integrate existing agents to achieve significant performance gains—***aligns with the paradigm of Agentic AI***. Similar to foundational works like ReAct [1] and Cursor [3], as well as advanced systems like STORM and OpenAI Deep Research [5, 6], the core value here lies in the orchestration of components to solve complex tasks unattainable by base models alone.

### **2. Technical Details & Pipeline Robustness *(reviewers vNNp, 7bLz)***

- **Structured Visual Coherence**: Unlike random compositing, our pipeline is ***layout-conditioned and role-aware***, grounded in 800K real-world layouts with shared global prompts for stylistic unity. Furthermore, our fine-tuned ART+ model (building on ART [7]) explicitly learns physical harmonization (e.g., lighting), ensuring superior global consistency.
- **Rigorous Optimization vs. Simple Blending**:
  - **LayerFLUX (Component)**: Rigorously optimized and training-free, it outperforms LayerDiffuse [8] with a 63.1% win rate in layer quality.
  - **MultiLayerFLUX (System)**: Adopts a bottom-up, aspect-ratio-preserving strategy respecting semantic roles. Unlike top-down methods (e.g., LayerDiff [9]), it enables ***scalable dataset bootstrapping from any SOTA generator.***
- **Model-Agnostic Design**: Validated using Qwen-Image [10] (Appendix), confirming our pipeline effectively transfers to foundation models beyond FLUX.

---

### Meta-Review · Area_Chair_DaFL · 2026-01-07

**Summary:**

The reviewers broadly agree that the paper addresses an important and underexplored problem by releasing a large-scale, open dataset suite for multi-layer transparent image generation, and they recognize the potential community value of such a resource. At the same time, several substantive concerns informed the reviewers’ overall assessments.

A primary concern across reviews is the limited realism and heavy reliance on synthetic, design-centric data. Multiple reviewers noted that even the “real” subset appears stylized or artificial, is small in scale, and does not convincingly capture the complexity of real-world multi-layer image composition. While the authors clarified that the dataset targets graphic design applications, reviewers remained uncertain about the role and practical usefulness of the real-image subset, as well as the extent to which the dataset supports multi-layer generation beyond design-focused scenarios.

Reviewers also raised questions regarding the methodological clarity and novelty of the proposed pipeline. In particular, it was unclear how multiple layers are enforced to belong to a coherent scene, how overlapping layers are handled, and whether the composition process goes beyond alpha blending guided by prompting and layout constraints. Some reviewers therefore, viewed the contribution as closer to a well-engineered dataset and system release.

From the AC perspective, these two concerns are not necessarily disqualifying: narrowing the scope to an important subfield (graphic design) is acceptable, and for a dataset/benchmark paper, strong algorithmic novelty is not a strict requirement. **However, rigorous and convincing analysis and evaluation of the proposed data construction and generation pipeline remain critical, and the current evaluation is viewed as limited.**

Specifically, reviewers noted the lack of clarity of the presentation, lack of explicit evaluation of text rendering quality, limited analysis of dataset, and insufficient evidence for physical plausibility across layers (e.g., transparency). In addition, the pipeline and benchmarks are perceived as largely FLUX-centric, with only limited evidence of generalization to other backbone models. The AC also raises new concerns regarding the evaluation and grounding of the TIPS score, the reliance on synthesized data rather than crawled layered data, and the overall dataset quality, particularly in terms of alpha matte accuracy and intrinsic transparency.

Overall, while reviewers acknowledged the dataset’s scale and careful construction, the insufficient evaluation and remaining ambiguity around dataset prevent the paper from meeting the acceptance bar for ICLR.

**Reviewer Concerns:**

## Solved

### Double-Blind Violation
- **Reviewer H3ue:** Public Hugging Face link on page 1 may reveal author identity.
- **Resolution:** Authors clarified that there is no identity information in the HuggingFace link and removed the link.

### Explanation of Table 1 Quality Assessment
- **Reviewer 7bLz:** Asked how “alpha quality” and “aesthetic” in Table 1 were assessed.
- **Resolution:** Authors explained the evaluation process (human inspection + categorical ratings).


### Dataset realism
- Reviewers 7bLz, vNNp, H3ue, and uqof questioned the realism of the dataset.
- **Resolution:** *Partially solved*. The authors clarified that the dataset is intentionally **design-centric**, targeting graphic design and editable layered content rather than physically accurate photo compositing. However, why the real dataset is regarded as more realistic is unclear, and the real dataset was explained **bridging set**, but its function is not clearly defined.




### Pipeline Novelty
- **Reviewer 7bLz / vNNp:** Concerned about the novelty of MultiLayerFLUX.
- **Resolution:** Authors added explanations on the contribution of layout conditioning, semantic roles, z-ordering, and artifact filtering, and argued that this paper is a dataset track paper
- **Status:** *Partially solved*, improved clarity, but novelty and guarantees remain debatable.



### Partial Response on FLUX only
- **Reviewer H3ue:** Pipeline and benchmarks are just based on FLUX
- **Resolution:** Authors added qualitative examples using Qwen-Image-Edit.
- **Status:** *Partially solved*, demonstrates feasibility, but lacks quantitative comparison.




### Open-Sourcing Pipeline and Benchmarks
- **Reviewer vNNp:** Asked whether datasets, pipeline, and evaluation tools will be released.
- Authors promised release
- **Status:** *Partially solved*




### Writing and Presentation Improvements
- **Reviewer 7bLz / H3ue:** Paper organization and figure readability issues.
- **Resolution:** Captions rewritten; related work expanded.
- **Status:** *Partially solved.*

---

## Issues Not Solved

### Use of Crawled Layered Data (from AC)
- It remains unclear why crawled layered images are **not used directly nor released**, and only layouts/prompts are extracted. Authors cite aesthetic degradation would affect the model performance, but provide **no empirical evidence or ablation**. Besides, the possibility of filtering crawled data using the same aesthetic/human pipeline is not explored.


### Alpha Matte Accuracy and Intrinsic Transparency (from AC)
- Alpha mattes are generated using **pre-trained matting models**, yet are claimed as accurate. Figure 18 shows poor handling of intrinsic transparency (e.g., semi-transparent interiors). In fact, the hard nut could be cracked in other ways, like human annotation.



### Evaluation of the  TIPS score (from AC)
- The  TIPS score was used as the metric; however is not evaluated.



### Text Rendering Evaluation
- **Reviewer uqof:** Text rendering accuracy is important, but not explicitly evaluated.
- No OCR-based or dedicated text accuracy analysis is provided.



### Physical Plausibility Across Layers
- **Reviewer vNNp:** Shadows, lighting, reflections, and physical interactions are not modeled.
- Authors argue this is also unsolved by prior work and outside scope.
- From an AC perspective, transparent objects are central and cannot be fully neglected.
- **Status:** *Not solved.*



### ART vs. ART+ Comparison Clarity
- **Reviewer 7bLz:** Differences are unclear; prompts should be shown.
- Authors promise to add prompts and zoom-ins.
- **No such revision appears in the current version.**
- **Status:** *Not solved.*



### Dataset Statistics
- **Reviewer 7bLz:** Requested dataset statistics on transparent object categories.
- Authors promised to add them. but **No evidence of inclusion in current revision.**
- **Status:** *Not solved.*

**Reviewer Scores:**

**Reviewer uqof (6 $\rightarrow$ 6)** On the one hand, the authors’ clarifications on dataset scope (design-centric focus) addressed the concern about dataset realism. On the other hand, missing a dedicated text-rendering evaluation prevents a score increase.

**Reviewer 7bLz (4$\rightarrow$ 4/6)** Authors provided additional explanations and analyses and explicitly engaged with the reviewer’s concerns.

**Reviewer vNNp ($\rightarrow$ 4)**  As core concerns about lack of naturalness, physical plausibility, and the absence of a principled approach to real-world transparent image acquisition remain largely unresolved despite clarifications.

**Reviewer H3ue ($\rightarrow$ 4)**  Since the reviewer explicitly stated that their core concerns --- particularly the style bias of graphic/cartoon, limited real-image diversity, and FLUX-centric methodology --- were not resolved by the rebuttal.

---

### Decision · Program_Chairs · 2026-01-26

Reject